# The circadian cryptochrome, CRY1, is a pro-tumorigenic factor that rhythmically modulates DNA repair

Ayesha A. Shafi [1], Chris M. McNair[1,2], Jennifer J. McCann[1,3], Mohammed Alshalalfa [4], Anton Shostak[5], Tesa M. Severson[6], Yanyun Zhu[6], Andre Bergman [6], Nicolas Gordon[1], Amy C. Mandigo [1], Saswati N. Chand[1], Peter Gallagher[1], Emanuela Dylgjeri [1], Talya S. Laufer[1], Irina A. Vasilevskaya[1], Matthew J. Schiewer[1,7], Michael Brunner[5], Felix Y. Feng [4], Wilbert Zwart [6] & Karen E. Knudsen [1,2,7 ✉]

Mechanisms regulating DNA repair processes remain incompletely defined. Here, the circadian factor CRY1, an evolutionarily conserved transcriptional coregulator, is identified as a tumor specific regulator of DNA repair. Key findings demonstrate that CRY1 expression is androgen-responsive and associates with poor outcome in prostate cancer. Functional studies and first-in-field mapping of the CRY1 cistrome and transcriptome reveal that CRY1 regulates DNA repair and the G2/M transition. DNA damage stabilizes CRY1 in cancer (in vitro, in vivo, and human tumors ex vivo), which proves critical for efficient DNA repair. Further mechanistic investigation shows that stabilized CRY1 temporally regulates expression of genes required for homologous recombination. Collectively, these findings reveal that CRY1 is hormone-induced in tumors, is further stabilized by genomic insult, and promotes DNA repair and cell survival through temporal transcriptional regulation. These studies identify the circadian factor CRY1 as pro-tumorigenic and nominate CRY1 as a new therapeutic target.

[1] Department of Cancer Biology, Thomas Jefferson University, Philadelphia, PA 19107, USA. [2] Sidney Kimmel Cancer Center, Thomas Jefferson University, Philadelphia, PA 19107, USA. [3] Duke University, Durham, NC 27708, USA. [4] Department of Radiation Oncology, University of California at San Francisco, San Francisco, CA 94115, USA. [5] Biochemistry Center, University of Heidelberg, Heidelberg, Germany. [6] Division of Oncogenomics, Oncode Institute, The Netherlands Cancer Institute, Amsterdam, The Netherlands. [7] Department of Urology, Medical Oncology and Radiation Oncology, Thomas Jefferson University, Philadelphia, PA, USA. ✉email: karen.knudsen@jefferson.edu

Prostate cancer (PCa) is the second leading cause of cancer death in US men[1]. First-line therapy for patients with disseminated disease targets the androgen receptor (AR), a ligand-dependent transcription factor required for PCa development and progression. Although androgen depletion and AR-targeted therapies are initially effective, recurrent castration-resistant prostate cancer (CRPC) arises for which there is no durable cure[2]. Notably, CRPC remains largely AR-dependent due to aberrant reactivation of AR through multiple distinct mechanisms that promote AR-mediated cell proliferation, DNA repair, and tumor survival[3]. Thus, there is an urgent need to develop novel strategies that enhance and/or act in concert with AR-targeted therapy.

Toward this end, recent findings highlight the potential of understanding and leveraging AR-dependent DNA repair. Clinical AR suppression has long been utilized for radiosensitization, and affords significant clinical benefit compared to radiation alone[4]. Subsequent mechanistic investigation revealed AR regulates DNA repair factor expression (including DNAPK < encoded by *PRKDC* >, Ku70/80, and PARP1)[5–8]. Functional studies identified AR-mediated DNAPK expression as required for AR-dependent double-strand break (DSB) repair, and identified additional functions downstream of DNAPK important for the metastatic process, forming the basis of an ongoing clinical trial (NCT02833883). Furthermore, a recent clinical study showed that tumors with pathogenic DNA repair factor alterations responded more favorably than those with no detectable DNA repair alterations to androgen depleting strategies[9]. Given the impact of AR-dependent DNA repair processes on PCa progression, further delineating the means by which AR regulates break resolution is of translational importance.

Here, analyses of advanced PCa unexpectedly identified CRY1 (cryptochrome 1), a transcriptional coregulator associated with the circadian clock[10] as a tumor specific, AR-mediated, critical effector of DNA repair that is deregulated in metastatic PCa patients and associated with poor outcome. Several epidemiological studies indicate that disruptions in circadian rhythm, such as jet lag, shift work, sleep disruption, and suppression of melatonin by exposure to light at night are all associated with increased risk of PCa, breast cancer, and colon cancer[4–8]. Loss of circadian control is also associated with poor efficacy of anticancer treatments and early mortality among cancer patients[9]. First-in-field CRY1 cistrome and transcriptome mapping identified CRY1 as a regulator of cell proliferation and DNA repair processes, which was functionally confirmed in across PCa model systems. To assess relevance, exogenous challenge with genotoxic stress (utilizing in vitro systems, in vivo models, and human tumors ex vivo) revealed that AR-induced *CRY1* is further stabilized by genomic insult, after which CRY1 binds to promoters of homologous recombination (HR) factors to regulate HR-mediated DNA damage response (DDR) in a cascading, temporal fashion inducing first the sensors/mediators of HR followed by induction of HR effectors. Congruently, CRY1 strongly correlated with HR gene expression in PCa. These collective findings reveal that androgen-regulated CRY1 is stabilized in response to genotoxic insult and governs rapid repair of DNA DSBs by directly regulating HR gene expression, thus modulating genome integrity and promoting CRPC growth. In sum, these studies identify a novel, tumor-specific mechanism by which hormones regulate DNA repair and are the first to delineate the molecular framework used by CRY1 in PCa progression. Thus, these findings identify CRY1 function as protumorigenic and nominate a new, targetable pathway for managing advanced PCa.

## Results

### CRY1 is induced by androgens and associated with poor outcome.
Recent studies underscore the importance of AR-mediated DNA repair factor regulation in PCa, yet this critical facet of AR signaling is incompletely defined. AR regulates a vast transcriptional network in response to androgen stimulation, as demonstrated by cistrome and transcriptome mapping, and AR-dependent DNA repair factor regulation is a major effector of the response to DNA damage[5–8]. Leveraging recent insight into genome-wide AR activity revealed that in the presence of androgen-stimulation AR binds multiple regions within the *CRY1* locus, encoding a transcriptional coregulator most well studied in circadian regulation[11–14] (Fig. 1a, Supplementary Fig. 1a). AR binding at the *CRY1* locus was conserved across the cell cycle in PCa with no significant change in mRNA expression (Supplementary Fig. 1a–b)[15], indicating that these events are of relevance in mitotically active cancer cells. Further investigation showed that CRY1 is also induced in response to androgen stimulation in CRPC cells (Fig. 1b), thus nominating CRY1 as an AR-regulated gene of putative relevance to advanced disease. Strikingly, AR binding to the *CRY1* locus was enriched in both newly assessed and archived analyses of PCa versus non-neoplastic tissues (Fig. 1c)[16]. Together, these observations nominate *CRY1* as a tumor specific, AR-regulated target gene.

Circadian factors including CRY1 hold diverse functions across multiple cancer types, with underpinning mechanisms poorly defined[10]. CRY1 harbors known functions in lower eukaryotes as an effector of both transcriptional control and resolution of UV-induced DNA damage; this latter function manifests through functional domains conferring photolyase activity with ~40–60% similarity of protein structure to evolutionarily conserved photolyases[10,17,18]. While *CRY1* somatic alterations and gene amplifications occur in several tumor types, (Supplementary Fig. 1c), *CRY1* is frequently amplified in PCa and CRPC (Supplementary Fig. 1c–e) to a similar frequency of *AR* and key target genes (*KLK3*, *FKBP5*) previously known to be amplified in PCa (Supplementary Fig. 1f). Moreover, CRY1 expression is strongly associated with metastasis and poor outcome (Fig. 1d, Supplementary Fig. 1g). Interestingly, even when adjusting for age, high CRY1 remained an independent prognostic variable (HR 1.56 with a 95% CI [1.04–2.34], $p = 0.029$] (Fig. 1d), further strengthening the importance of CRY1 functioning as a protumorigenic factor in PCa. Moreover, the link between CRY1 and aggressive disease was assessed in a large cohort of radical prostatectomy specimens ($n = 5239$), wherein CRY1 positively correlated with increased genomic risk of metastasis with a correlation coefficient of 0.07 ($p = 1e^{-7}$) (Fig. 1e, Supplementary Fig. 1g). The genomic risk of metastasis is defined based on the Decipher test score which is a strong predictor of metastasis. Combined, these studies implicate CRY1 induction (as achieved by AR signaling and/or amplification of the *CRY1* locus), as an effector of disease progression.

### CRY1 cistrome mapping reveals interplay with oncogenic transcription factors.
Given the identification of CRY1 as an androgen-regulated gene of cancer relevance, the mechanism through which CRY1 influences cancer outcomes was assessed. Genome-wide understanding of CRY1 function is poorly understood, and in humans has been limited to a single osteosarcoma model[19]. CRY1 cistrome mapping in CRPC using a stringent cutoff identified 2551 CRY1-bound sites (Fig. 2a, left). As expected, CRY1 bound to regulatory regions encoding core circadian clock machinery, including *PER3*, *PER2*, *CRY2*, and *CRY1* itself (Fig. 2a, right), with the majority of total CRY1 binding events occurring at proximal promoters (37.9%) or intronic regions (32.6%) (Fig. 2b). Notably, binding to genes governing circadian function represented a minute fraction (2.5%) of CRY1 binding events (Supplementary Fig. 2a-b),

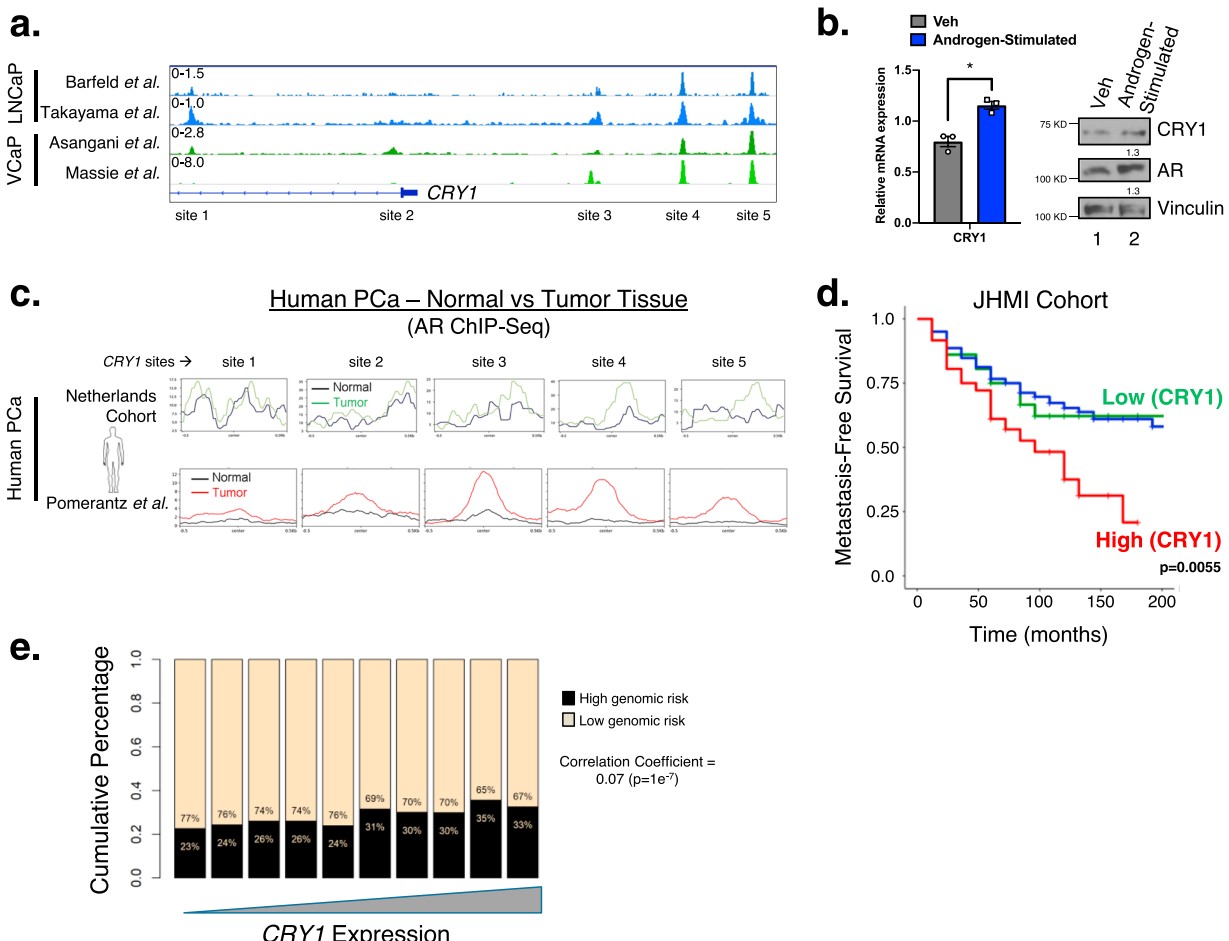

**Fig. 1 CRY1 is induced by androgens and associated with poor outcome. a** AR binding sites on CRY1 in several PCa datasets, including in Barfeld et al. 2017, Takayama et al. 2018, Asangani et al. 2014, and Massie et al. 2011. Genomic traces showing AR binding sites on CRY1. **b** C4-2 cells in hormone-deprived media (Veh) for 48 h and treated with 10 nM DHT for 16 h (Androgen-Stimulated). Cells were harvested for RNA (CRY1 and 18 S) and protein (CRY1, AR, and Vinculin). **c** AR binding sites on CRY1 in human tissue comparing normal and tumor samples from a cohort of PCa patients in the Netherlands from Dr. Wilbert Zwart and Pomerantz et al. 2015 dataset. **d** In the JHMI retrospective cohort, patient samples were split into groups based on CRY1 expression into top 10% vs middle 80% vs low 10%. Patients with high (top 10%) CRY1 expression are associated with poor metastatic outcome. **e** CRY1 expression was compared to the high Decipher 5239 prospective radical prostatectomy samples. $N = 3$ independent experiments. Data are presented as mean values ± SEM and analyzed using Student's t-test (*$p < 0.05$). Statistical significance was evaluated at 0.05 alpha level with GraphPadPrism, version 8.3.1, Mac. Source data are provided in the Source Data file.

indicating that CRY1 holds functions in PCa distinct from circadian gene regulation. This postulate was further underscored by the failure of PCa models to display circadian rhythm patterns of gene networks. These data are the first to demonstrate that in human cells CRY1 functions beyond circadian control and significantly extends previous murine models, which revealed that CRY1 functions distinctly from other circadian clock repressors (i.e., PER1, PER2, and CRY2) and exhibits a binding pattern distinct from circadian clock activators (i.e., CLOCK and BMAL1)[20–22]. Given this new knowledge, it was critical to assess the non-circadian functions of CRY1 in cancer cells.

To identify CRY1 functions outside of circadian regulation, analyses of CRY1 binding sites adjacent to transcriptional start site (TSS) revealed enrichment for growth factor signaling, DNA repair, and metabolic processes (Supplementary Fig. 2c), in addition to expected circadian networks. To investigate potential mechanisms of CRY1 function, de novo motif analysis was assessed using a window of 50 bp adjacent to the center of binding (Fig. 2c). In addition to motifs for NPAS (a factor known to interact with CRY1)[10,23], multiple motifs of PCa relevance

were enriched proximal to CRY1 binding, including several factors elevated in PCa, linked with androgen-associated cancer growth, and involved in the epithelial-mesenchymal transition and cancer progression (e.g., FOXD2, SP2)[24–27]. The concept that CRY1 binding is enriched for motifs associated with malignant progression was further substantiated through known motif analysis using a broad window around the center of binding (500 bp) (Fig. 2d, Supplementary Fig. 2d–e), wherein expected enrichment of circadian regulated factors (CLOCK, BMAL1, E-Box) was complemented by enrichment of cancer-associated transcription factors of PCa relevance, including several with oncogenic activity (e.g., c-Myc, Max, USF1, SP1, ETS factors, HIF-1α)[28–34]. Additional enriched motifs correspond to components of the FoxA1/AR complex or are directly driven by AR to promote AR signaling, cell proliferation/invasion, metabolic rewiring, and ultimately tumor growth (Fig. 2d, Supplementary Fig. 2d–e)[35]. These observed enrichments and related pathway analyses (Supplementary Fig. 2c) provide the first insight into genome-wide CRY1 activity in adenocarcinomas and reveal functions beyond circadian regulation linked to oncogenic factors.

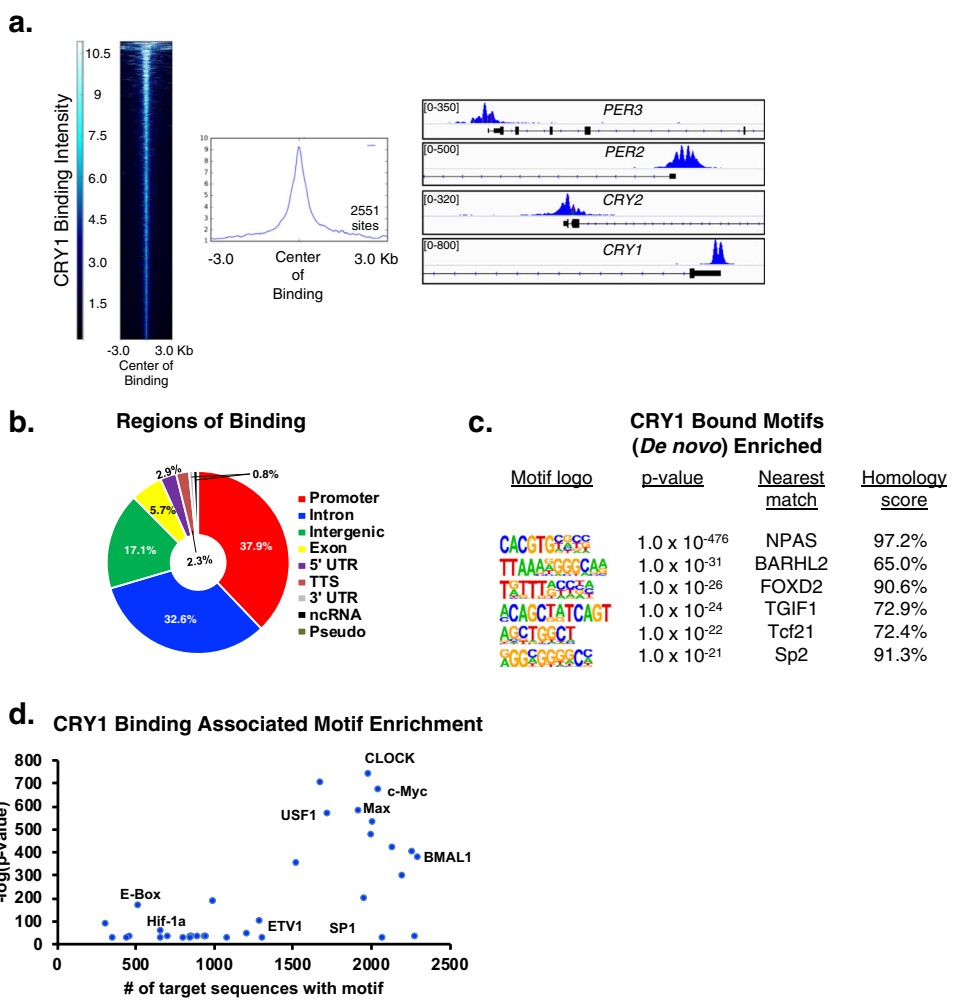

**Fig. 2 CRY1 cistrome mapping reveals interplay with oncogenic transcription factors. a** Heatmap and profile plots showing number and intensity of binding sites. Right—Genomic traces of CRY1 binding at core circadian clock genes. **b** Genomic distribution analysis representing genomic binding regions. Percent of total binding is shown in indicated regions. **c** De novo motif enrichment of CRY1 binding in vehicle treated C4-2 cells within 50 bp binding window on each side of the center of binding with the cutoff of $p < 10^{-20}$. **d** Known motif enrichments of CRY1 binding in vehicle treated C4-2 cells within 500 bp binding window on each side of the center of binding with the cutoff of $p < 10^{-20}$.

**CRY1 governs pathways critical for cell cycle regulation and response to DNA damage.** The observation that CRY1 is associated with poor outcome in metastatic PCa, coupled with linkage of the newly identified CRY1 cistrome to cancer-promoting pathways, suggests that CRY1 plays roles distinct from circadian regulation in PCa. To challenge this hypothesis, genome-wide assessment of the CRY1-sensitive transcriptional networks were evaluated utilizing newly generated doxycycline-regulated isogenic paired model systems of inducible *CRY1* knockdown. As shown, ~70% ablation of CRY1 protein was achieved after shRNA induction (Fig. 3a, left), subsequent to which the first whole-transcriptome analysis of CRY1 was assessed in human carcinomas (Fig. 3a, right), with strong consistency amongst biological replicates (Supplementary Fig. 3a). Major transcriptional changes were observed (2416 upregulated, 2736 downregulated genes) after CRY1 knockdown (adjusted *p*-value < 0.05) indicating that CRY1 influences large gene networks. Gene set enrichment analysis (GSEA) (Fig. 3b, Supplementary Fig. 3b) revealed that CRY1 governs transcriptional programs of cancer relevance, including those involved in DNA replication, cell cycle regulation, and multiple DNA repair processes. The biological effect of CRY1 on cell cycle progression in CRPC was assessed in multiple distinct isogenic pairs, wherein CRY1 ablation resulted in G2/M

arrest (Fig. 3c). Complementary studies were also performed using a well-studied activator of CRY1 (KL001), which functions by stabilizing CRY1 and preventing ubiquitin degradation[36]. As shown, KL001 increased the steady state of CRY1 in all tested CRPC model systems but did not result in measurable changes in cell cycle (Fig. 3d). While these studies reveal a requirement of CRY1 for cell cycle progression, heightened CRY1 protein proved insufficient to alter cell cycle position. Thus, tumor-associated CRY1 appears to be essential for cellular proliferation but alone is insufficient to drive a hyperproliferative phenotype.

Conversely, investigation of transcriptome linkages to DNA repair response revealed robust phenotypes of cancer relevance. Given the identification of CRY1 as an AR-regulated gene (Fig. 1), combined with the recent discoveries that AR promotes DSB repair and the robust preclinical and clinical data demonstrating that suppression of AR activity compromises DNA repair and confers radiosensitization, interrogation of putative CRY1-mediated DNA repair gene networks were prioritized. Numerous DNA repair factor networks proved sensitive to CRY1 depletion, including those essential for UV response, mismatch repair (MMR), nucleotide excision repair (NER), and HR (Fig. 3b, Supplementary Fig. 3c). These findings were unexpected, as the only DNA repair process previously linked to circadian clock

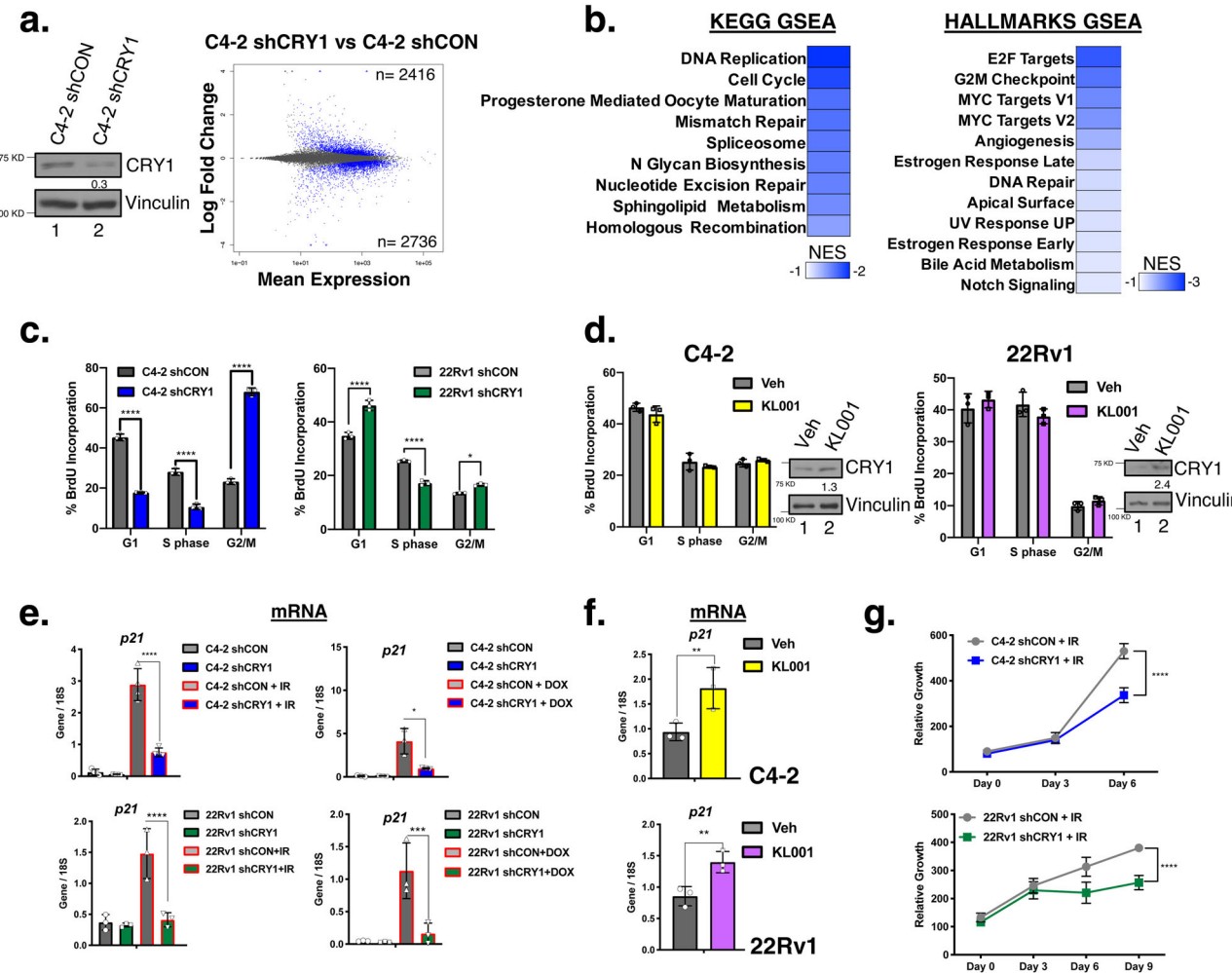

**Fig. 3 The CRY1-sensitive transcriptome identifies alterations in DNA repair processes. a**, **c**, **e** CRY1 expression was knocked down in C4-2-shCRY1 cells for 72 h. **a** RNA-Seq analysis was performed on quadruplet samples for C4-2-shCON and C4-2-shCRY1 cells. MA plot depicts gene expression modulation with the number of significant transcripts upregulated (top) and downregulated (bottom) in blue. **b** GSEA of RNA-Seq (KEGG and HALLMARKS Pathways) identified enriched and deenriched pathways for CRY1-regulated pathways using FDR < 0.25. **c** Flow analysis was performed. **d** C4-2 and 22Rv1 cells were treated with 10 μM KL001 at Day 0 and harvested for flow analysis. **e** After CRY1 was knocked down, cells were treated with 5 Gy IR for 4 or 24 h. **f** C4-2 and 22Rv1 cells were treated with 10 μM KL001 for 6 and 24 h, respectively. **g** C4-2-shCRY1 and 22Rv1-shCRY1 cells were treated with 5 Gy IR at Day 0 and harvested at indicated days for Pico Green to assess relative growth. $N = 3$ independent experiments. Data are presented as mean values ± SEM and analyzed using two-way Anova (*$p < 0.05$, **$p < 0.01$, ***$p < 0.001$, and ****$p < 0.0001$). Statistical significance was evaluated at 0.05 alpha level with GraphPadPrism, version 8.3.1, Mac. Source data are provided in the Source Data file.

factors is NER[10,37,38]. To evaluate the functional consequence of CRY1 on other DNA repair effects, genomic insults conferring DSB (ionizing radiation, IR or Doxorubicin, DOX) were utilized to examine changes in $p21^{cip1}$ (*CDKN1A*) mRNA after CRY1 perturbation as a readout of DNA damage cell cycle checkpoint induction (Fig. 3e)[3,10]. As expected, IR and DOX increased p21[cip1] expression in control cells of multiple CRPC backgrounds with intact CRY1 (C4-2 shCON, 22Rv1 shCON), but this DNA damage-induced event was significantly diminished in the absence of CRY1 (60–75% reduction in the isogenic models). Similar results were seen with transient transfection of siRNA targeting CRY1 in C4-2 and 22Rv1 cells. Conversely, CRY1 stabilization via KL001 resulted in modest upregulation of p21[cip1] levels even in untreated cells, with a 1.5–2-fold further induction after CRY1 activation (Fig. 3f). These data identify CRY1 as required and sufficient to induce cell cycle checkpoint induction after DSB. Downstream analyses demonstrated the biological relevance, as evidenced by a 1.6–1.8-fold reduction in surviving cells after IR in CRY1-depleted cells (Fig. 3g) and

maintained cell growth after DNA damage in CRY1 activated cells (Supplementary Fig. 3d). These findings reveal new insight into CRY1 function as a modulator of cell cycle checkpoint control and cell proliferation in response to DSB, and implicate CRY1 in promoting cancer cell survival.

**DNA damage stabilizes CRY1.** Whereas CRY1 has been well studied as a transcriptional coregulator important for regulation of circadian rhythm, the protein is highly homologous (~40–60% similarity) to evolutionarily conserved photolyases that are stabilized in the presence of UV-induced DNA damage in plants and *Drosophila*, and promote resolution of thymidine dimers[17,33,34]. As a role for CRY1 in DNA repair in mammalian cells or after DSB was unknown, the impact of genomic insult on CRY1 function was evaluated in CRPC. CRY1 protein levels increased as a function of escalating doses of IR (Fig. 4a), with a 2–3-fold induction observed 4-h post-IR, indicating that the effects of DNA damage on CRY1 accumulation are rapid. A time dependent increase in CRY1 was also observed from 0–8 h after 5 Gy

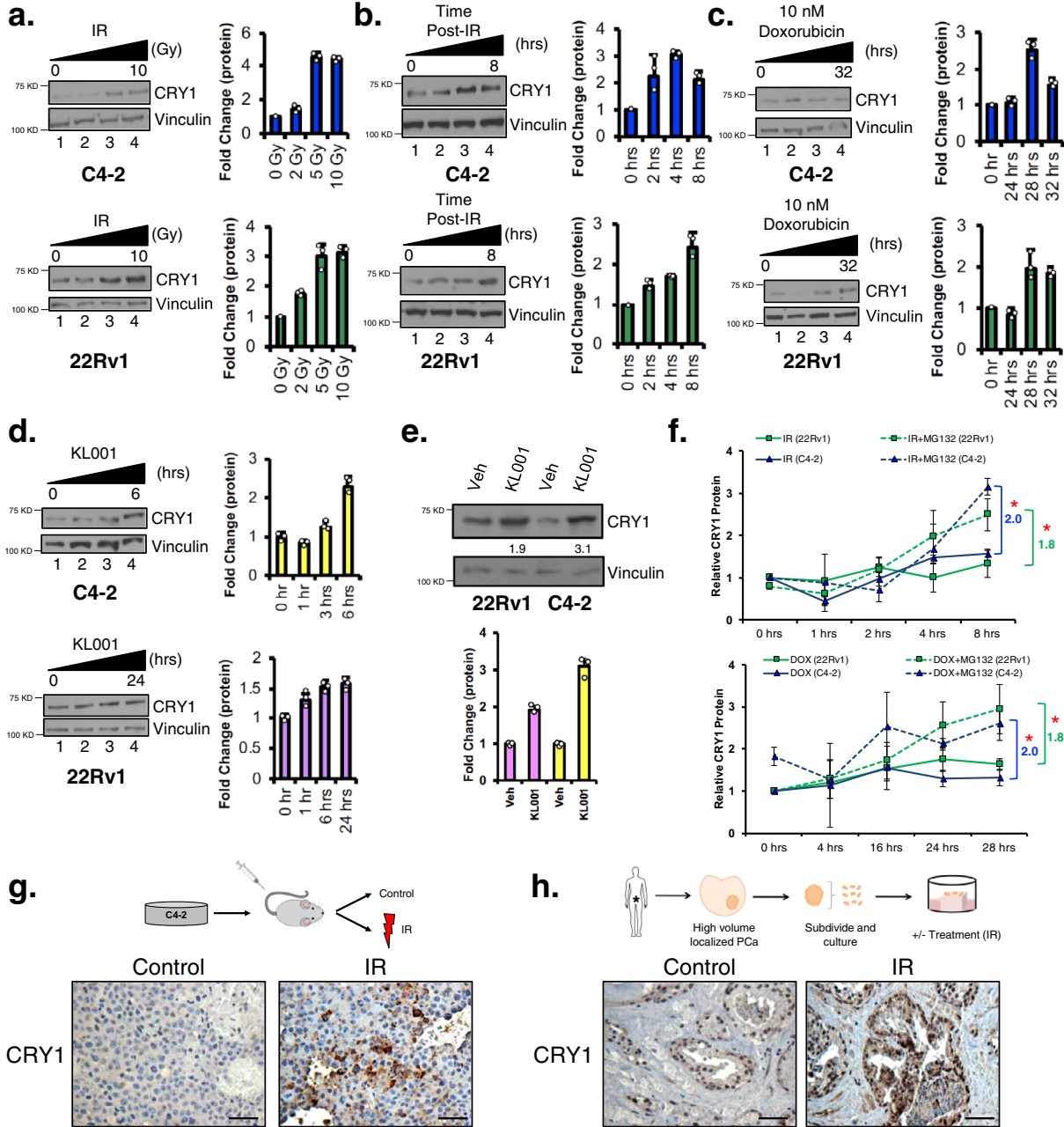

**Fig. 4 DNA damage results in CRY1 stabilization. a** C4-2 and 22Rv1 cells were treated with increasing doses of IR for 4 h. **b** C4-2 and 22Rv1 cells were treated with 5 Gy IR for the indicated times. **c** C4-2 and 22Rv1 cells were treated with 10 nM Doxorubicin (DOX). **d** C4-2 and 22Rv1 cells were treated with 10 μM KL001. **e** C4-2 and 22Rv1 cells were treated with 10 μM KL001 for 6 and 24 h, respectively. **a–e** Cells were harvested; protein expression of CRY1 and Vinculin was analyzed and quantified using ImageJ. **f** C4-2 and 22Rv1 cells were treated with 1 μM MG132 and 5 Gy IR or pretreated with 1 μM MG132 for 4 h and then treated with 10 nM DOX. Cells were harvested; protein expression of CRY1 and Vinculin was analyzed using ImageJ (*$p < 0.05$ for last timepoint compared to their respective control without MG132 treatment). **g** C4-2 xenografts were treated with control or 4 Gy IR and harvested after 20–30 days for CRY1 IHC analysis. Scale bar 250 μm. **h** PDE PCa tissue was treated with 0.5 Gy IR. Tissue was harvested at 2 days for CRY1 IHC analysis. Scale bar 250 μm. $N = 3$ independent experiments. Data are presented as mean values ± SEM and analyzed using two-way Anova (*$p < 0.05$, **$p < 0.01$, ***$p < 0.001$, and ****$p < 0.0001$). Statistical significance was evaluated at 0.05 alpha level with GraphPadPrism, version 8.3.1, Mac. Source data are provided in the Source Data file.

IR, wherein protein levels peaked (Fig. 4b). Similar results were seen after DOX treatment (Fig. 4c). These observations demonstrate that CRY1 is rapidly enriched after genomic insult, suggesting mechanistic regulation of the protein by post-translational mechanisms.

The effects of DNA damage on CRY1 stability were further compared to those in cells treated with CRY1 activator KL001 in the absence of DNA damage. As shown (Fig. 4d), KL001 stabilized

CRY1 and caused a 1.5–2-fold increase in protein within 6 h, similar to the induction and timing observed after DNA damage (Supplementary Fig. 4a–b). Importantly, basal CRY1 protein expression was higher in 22Rv1 cells than C4-2 cells, which likely accounts for the larger fold increase in protein expression depicted in C4-2 cells (Fig. 4e). To discern if DNA damaged-induced CRY1 stabilization is also attributed to post-translational regulation, proteasomal degradation and half-life studies for

CRY1 were performed after genotoxic insult, using KL001 as comparison. As shown, use of proteasome inhibitor MG132, to prevent protein degradation and examine stability of CRY1 expression resulted in a significant (1.8–2 fold) increase of CRY1 in the presence of DNA damage (Fig. 4f, Supplementary Fig. 4c–d). Conversely, cycloheximide to examine the half-life of CRY1 did not result in any significant change after genotoxic insult (Supplementary Fig. 4e–f). These findings strongly support the conclusion that CRY1 is activated by stabilization after DNA damage, similar to that observed with the known CRY1 activator KL001.

To assess CRY1 regulation in vivo and gain insight into potential clinical relevance, subsequent studies examined the impact of DNA damage on CRY1 regulation in CRPC xenografts. As shown, tumors treated with 4 Gy IR have increased CRY1 protein expression (Fig. 4g). Complementary studies using primary ex vivo patient-derived explants of human PCa were performed as previously described[39–41] in the presence of 0.5 Gy IR for 48 h (Fig. 4h). Similar to in vitro and in vivo studies, CRY1 protein increased ex vivo in response to IR treatment in primary human tissue (Fig. 4h). Combined, multiple models of CRPC (in vitro, in vivo) and ex vivo irradiation of primary human tumors strongly indicate that a proximal response to DNA damage is CRY1 stabilization. Given the strong link between CRY1 and poor outcome in this disease type, these observations prompted deeper investigation into the molecular consequence of CRY1 function.

**CRY1 modulates DNA repair factor expression and homologous recombination**. Given the clinical observations that CRY1 is upregulated in PCa and associated with poor outcome (Fig. 1), combined with molecular observations that the protein induces gene networks of importance for DNA repair (Figs. 2, 3) and is stabilized after DNA damage (Fig. 4), computational strategies were employed to integrate cistrome and transcriptome analyses, with the goal of identifying the mechanism(s) of action by which CRY1 modulates the response to DNA damage. Initially using a guilty-by-association approach, the identified CRY1-responsive transcriptome (Fig. 3, defined as significantly ($p < 0.05$) altered genes after CRY1 knockdown) was integrated with the CRY1 cistrome (defined as genes with significant CRY1 binding within the TSS ($n = 1929$), Fig. 2a). This stringent analysis resulted in 750 genes likely to be directly regulated by CRY1. This dataset was subsequently compared to previously curated and published DNA repair pathways available on MSigDB (Supplementary Fig. 5a), which were grouped into known genes involved in HR, MMR, NER, BER (base excision repair), and NHEJ (non-homologous end joining) (Supplementary Fig. 5b) to establish the categories of DNA repair factors under CRY1 regulation (Fig. 5a). These analyses revealed widespread and potentially critical roles for CRY1 in regulating factors associated with HR (62% of CRY1-regulated genes), MMR (53%), BER (46%), NER (39%), and NHEJ (14%). Significant effects were associated with HR, suggesting that CRY1 is required for expression of major HR factors, including *RAD51, BRCA1, BRCA2, XRCC3*, and *CHEK1*. The capacity of CRY1 to alter HR gene expression was validated in additional CRPC models (Fig. 5b). These findings provide the first evidence that CRY1 modulates factors associated with HR, thus regulating response to DNA damage and enhances tumor aggressiveness through these mechanisms.

Given these observations, the impact of CRY1 on HR-mediated DNA repair was assessed in functional assays. Initially, a well-established mammalian reporter cell line (U2OS-DR-GFP) was used to measure HR efficiency[42–44], wherein HR is evidenced by restored GFP expression (Fig. 5c, top). As shown, CRY1 ablation

significantly decreased HR efficiency (3.9-fold reduction), similar to that observed after ATM inhibition (4.6-fold reduction) or BRCA1 knockdown (8.6-fold reduction) (Fig. 5c, Supplementary Fig. 5c). Conversely CRY1 activation (KL001) trended to an increase in HR efficiency. These findings nominate CRY1 as a positive effector of HR. To assess impact in PCa, CRY1 depletion or CRY1 stabilization (KL001) strategies were employed and cells were challenged by genotoxic insult (IR or DOX). As expected, *ATM* and *CHK2* mRNA increased in response to insult, but this induction was blunted after CRY1 knockdown; conversely, KL001 induced *ATM* and *CHK2* mRNA after DNA damage (Fig. 6a). Use of siRNA to target CRY1 in C4-2 and 22Rv1 cells resulted in similar outcomes (Supplementary Fig. 6a). Concordant changes in ATM and CHK2 protein were enhanced after insult-induced DNA damage and reduced as a result of CRY1 depletion (Fig. 6b, Supplementary Fig. 6b).

To assess the impact of CRY1-mediated HR factor regulation on DNA repair, resolution of DSBs after CRY1 manipulation was quantified via γH2AX and RAD51 foci (Fig. 6c, d, Supplementary Fig. 6c–e). As expected, γH2AX and RAD51 foci were induced rapidly after IR and resolved in control cells by 24 h. However, this process was markedly delayed in CRY1-depleted CRPC cells, consistent with impaired HR function. By contrast, CRY1 induction, as is observed in human tumors (Fig. 1), accelerated time to resolution of DNA breaks (Fig. 6d, Supplementary Fig. 6e). Together, these results reveal unexpected roles for CRY1 as a critical regulator of HR-mediated repair in CRPC, manifested through regulation of HR factor expression.

**CRY1 temporally modulates homologous recombination factor expression**. Given these findings, the mechanistic underpinnings of CRY1-mediated HR regulation were further considered. CRY1 did not colocalize with γH2AX foci, indicating that the role of CRY1 is likely upstream of repair (Supplementary Fig. 7a). As HR-mediated repair consists of sensors/mediators (MRE11 and RAD50), transducers (ATM), activators/adaptors (RAD51, XRCC3, and POLD2), and ultimately effectors (CHK2 and γH2AX) to resolve DSBs[3], the contribution of CRY1 to regulation of these factors was investigated. As expected, based on integration of cistrome and transcriptome data (Fig. 5), CRY1 knockdown decreased mRNA expression of *ATM*, *MRE11A*, *RAD50*, *RAD51*, *XRCC3*, and *POLD2* (Fig. 7a, Supplementary Fig. 7b). Congruently, ChIP-qPCR using binding sites informed by the cistrome mapping (Fig. 2) revealed increased CRY1 binding at DNA repair sensors at 30 min post-IR treatment, which was diminished after 4 h, suggestive that CRY1 function occurs rapidly after DNA damage (Fig. 7b, Supplementary Fig. 7c–d). Subsequently, downstream HR components (XRCC3 and POLD2) emerged at later timepoints (4-h post-IR treatment), thus revealing CRY1-mediated temporal control of HR factor expression. Identification of CRY1 as an effector of HR gene induction is further substantiated by the observation that CRY1 stabilization via KL001 resulted in increased ATM, MRE11A, RAD50, and XRCC3 expression, or conversely, significant attenuation after CRY1 depletion (Fig. 7c, Supplementary Fig. 7e). Interestingly, the binding of CRY1 to these key DNA repair factors (MRE11A, ATM, POLD2, and XRCC3) and another circadian component (CRY2) was not dependent on hormone stimulation (Supplementary Fig. 7f). CRY1 significantly bound to these sites in hormone depleted (CDT) and androgen treated (DHT) conditions, supporting an independent role of CRY1 as a protumorigenic factor. Clinically, a robust association between CRY1 and ATM, MRE11A, or RAD50 was observed in metastatic CRPC (Fig. 7d, Supplementary Fig. 7g), consistent with the evidence gleaned herein using a breadth of in vitro systems,

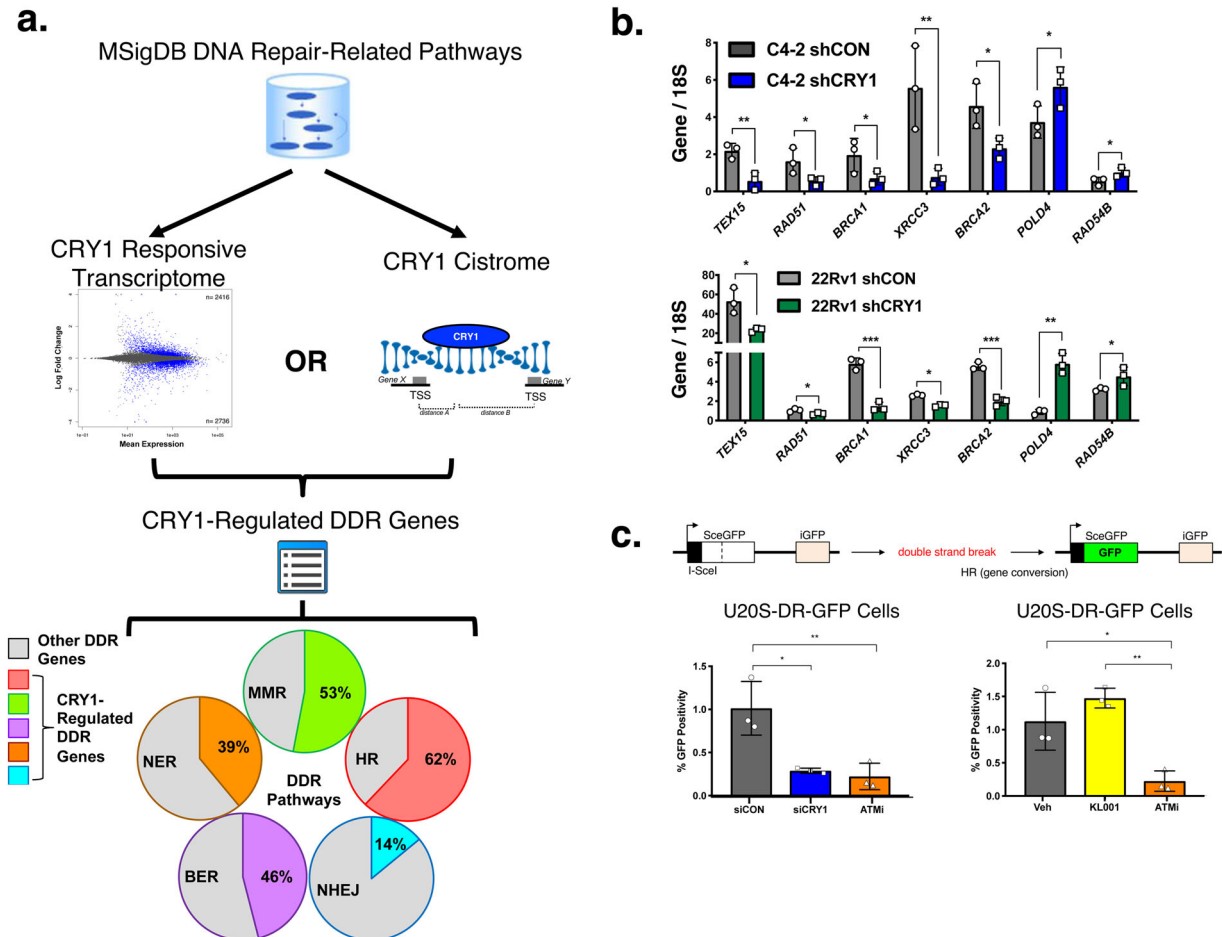

**Fig. 5 CRY1 transcriptome and cistrome analyses identify direct regulation of DNA repair by CRY1. a** Schematic describing the comparison of RNA-Seq and ChIP-Seq datasets. Briefly, the DNA damage response transcripts regulated by CRY1 with $p < 0.05$ (no fold change cutoff) and DDR genes with a CRY1 binding site within a TSS of binding were identified and organized into specific DDR pathways. **b** CRY1 expression was knocked down in C4-2-shCRY1 and 22Rv1-shCRY1 cells for 72 h, RNA was harvested, and qPCR was performed. **c** CRY1 was knocked down in U20S-DR-GFP cells for 72 h via siRNA and cells were treated with ATM inhibitor for 24 h or with 10 μM KL001 for 48 h. Cells were harvested for flow cytometry. $N = 3$ independent experiments. Data are presented as mean values ± SEM and analyzed using two-way Anova (*$p < 0.05$, **$p < 0.01$, and ***$p < 0.001$). Statistical significance was evaluated at 0.05 alpha level with GraphPadPrism, version 8.3.1, Mac. Source data are provided in the Source Data file.

in vivo xenografts, and primary human tumors. Combined, these findings identify CRY1 as a tumor specific, AR-induced effector of poor outcome in PCa and identify entirely new functions of CRY1 to temporally control HR and the response to genomic insult.

## Discussion

Recent observations that DNA repair alterations are highly prevalent in aggressive PCa and predict for poor outcome highlight an urgent need to discern the mechanisms that regulate the DNA repair process. Studies herein identify an unexpected, temporally modulated mechanism of DSB repair regulation, mediated by tumor-specific, AR-mediated induction of the CRY1 transcriptional regulator. Key findings demonstrate that: (i) CRY1 is induced by androgens in a tumor-specific manner; (ii) CRY1 induction is mediated in PCa by both AR and amplification of the *CRY1* locus; (iii) CRY1 cooperates with oncogenic transcription factors known to induce aggressive PCa; (iv) CRY1 governs pathways critical for cell cycle regulation and the response to DNA damage through transcriptional regulation; (v) DNA damage stabilizes CRY1 (vi) CRY1 is a direct and temporal modulator of DNA repair factor expression, initially inducing sensors of DSBs, mediators of HR, and downstream HR factors in

sequence; and (vii) CRY1 is strongly associated with HR factor expression and poor outcome in human disease. These studies are the first to delineate the molecular framework used by CRY1 to enhance cancer progression and nominate CRY1-DNA repair pathways as a potential node for therapeutic targeting in late stage disease.

The present study reveals the first genome-wide insight into CRY1 function in human carcinomas and provides the basis to discern the molecular underpinnings(s) by which CRY1 impacts cancer outcomes. A central finding is that CRY1 regulates gene expression far beyond that associated with circadian rhythm. Analysis of the PCa CRY1 cistrome revealed the prominence of CRY1 binding beyond genes governing circadian rhythm with only 49 of the 1929 (2.5%) CRY1-bound genes associated with circadian function (Supplementary Fig. 2a–b), further emphasizing the concept that CRY1 functions beyond the canonical circadian clock. This premise is supported by previous CRY1 analysis in circadian synchronized mouse livers, which demonstrated that CRY1 functions distinctly from other circadian clock repressors (e.g., PER1, PER2, and CRY2) and exhibits a binding pattern distinct from circadian clock activators (e.g., CLOCK and BMAL1)[20–22]. Moreover, in these previous mouse studies, CRY1 exhibited a unique pattern that peaked at circadian time zero (the

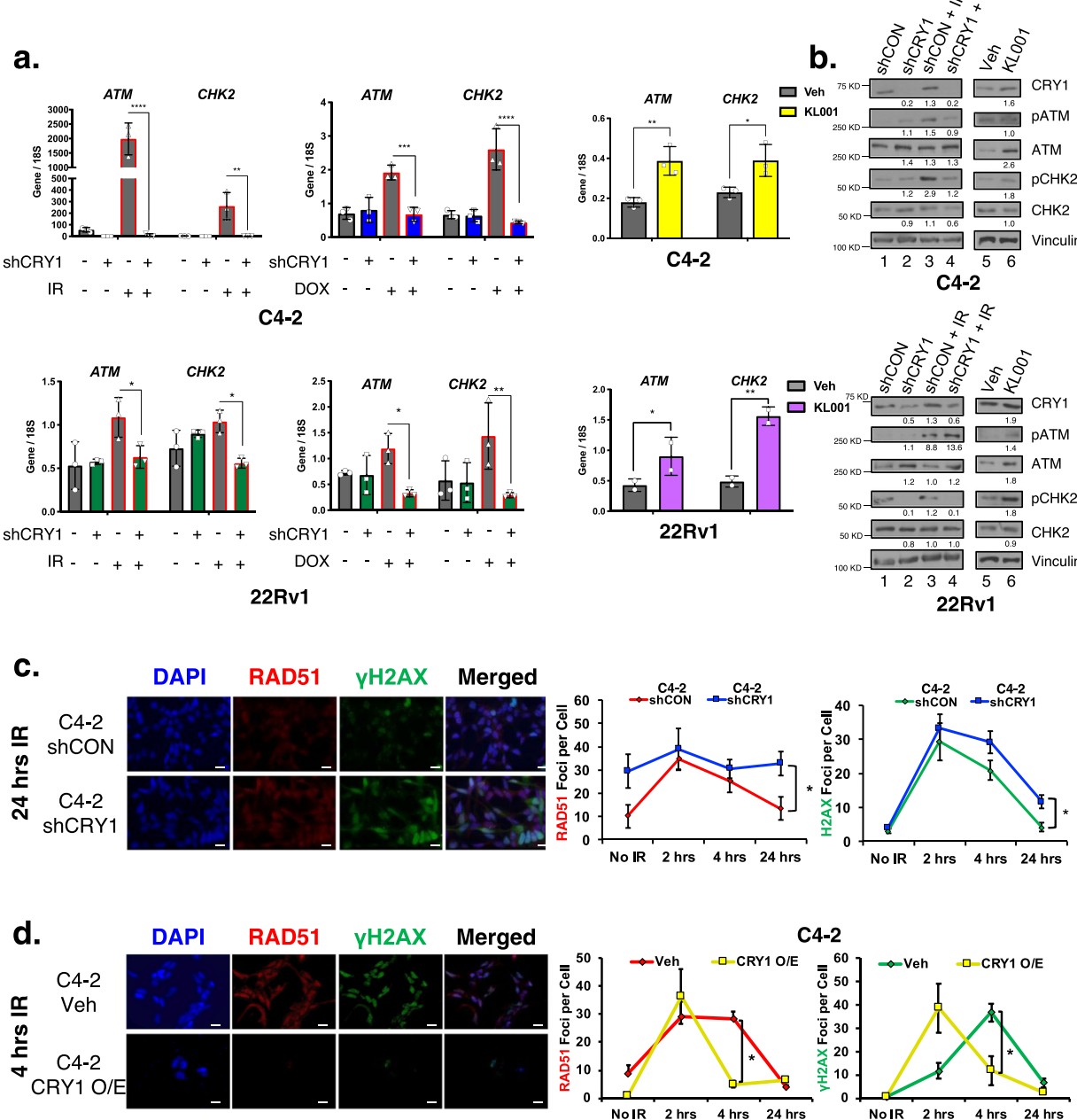

**Fig. 6 CRY1 promotes homologous recombination (HR)-mediated DNA damage response. a**, **b** CRY1 was knocked down in C4-2-shCRY1 and 22Rv1-shCRY1 cells for 72 h and cells were treated with 5 Gy IR for 4 or 24 h, respectively. C4-2 and 22Rv1 cells were treated with 10 µM KL001 for 6 and 24 h, respectively. Cells were harvested for RNA and protein. **c** CRY1 was knocked down in C4-2-shCRY1 and 22Rv1-shCRY1 cells for 72 h. **d** CRY1 expression was transiently overexpressed with transfection of a CRY1 plasmid for 48 h. **c**, **d** Cells were treated, fixed at the indicated timepoints, stained with γ-H2AX and RAD51 antibodies, and imaged by confocal microscopy. Foci were counted and plotted as foci per cell. Scale bar 250 µm. $N = 3$ independent experiments. Data are presented as mean values ± SEM and analyzed using Student's $t$-test, one-way Anova, or two-way Anova (*$p < 0.05$, **$p < 0.01$, ***$p < 0.001$, and ****$p < 0.0001$). Statistical significance was evaluated at 0.05 alpha level with GraphPadPrism, version 8.3.1, Mac. Source data are provided in the Source Data file.

onset of activity for diurnal organisms) unlike the other core circadian clock genes[45,46]. Furthermore, these previous circadian synced studies have reported that binding of individual circadian transcriptional factors do not correlate with the circadian regulator's binding phase[45,46]. Taken together, these observations strongly support the notion that CRY1 has its own binding pattern that extends beyond canonical circadian function. Intriguingly, these findings further support that even with the discovery of the lack of consistent circadian rhythm patterns in the PCa

model systems utilized for this study, assessment of the non-circadian functions of CRY1 is key to understanding drivers of disease progression. Subsequent studies indicated that CRY1 bound to several nuclear receptors independent of other clock proteins and functioned as co-repressors to PXR to mediate xenobiotic metabolism in liver and kidney cells[47]. Thus, observations in multiple tissues identify CRY1 activities beyond canonical circadian function, which have yet to be investigated in human carcinomas.

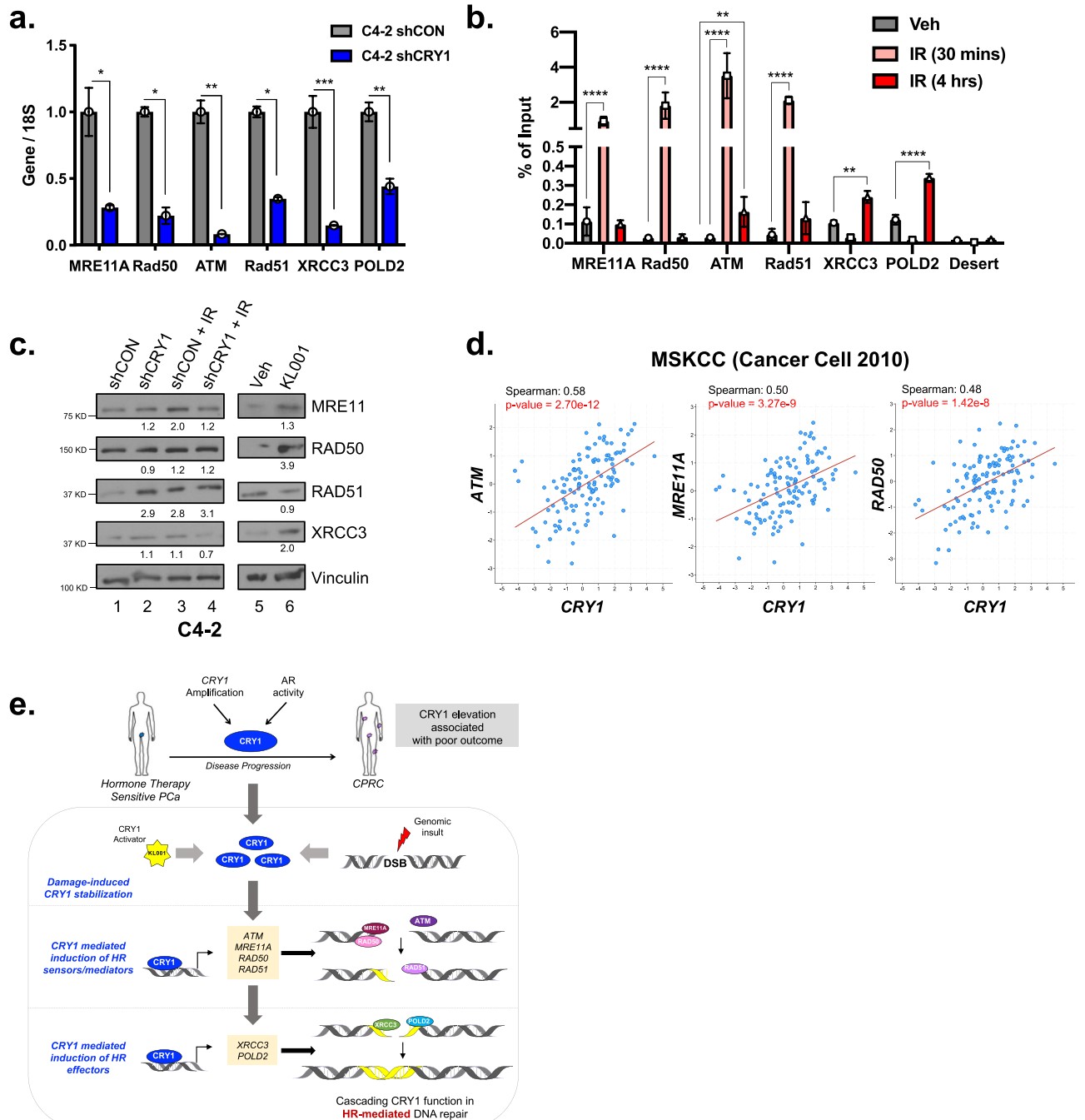

**Fig. 7 CRY1 regulates HR gene expression via binding to promoters and CRY1 correlates with HR gene expression in human disease. a** CRY1 was knocked down in C4-2-shCRY1 cells for 72 h. **b** ChIP-qPCR was performed to examine CRY1 binding after 30 mins and 4 h of 5 Gy IR treatment in C4-2 cells. Binding is plotted as percent of input. **c** CRY1 was knocked down in C4-2-shCRY1 cells for 72 h and cells were treated with 5 Gy IR for 4 h. C4-2 cells were treated with 10 μM KL001 for 6 h. **d** Co-expression of *CRY1* and either *ATM*, *MRE11A*, and *RAD50* mRNA in PCa tissue from publicly available data from MSKCC (Cancer Cell 2010). **e** Model summarizing the findings from this study. $N = 3$ independent experiments. Data are presented as mean values ± SEM and analyzed using Student's *t*-test or two-way Anova (*$p < 0.05$, **$p < 0.01$, ***$p < 0.001$, and ****$p < 0.0001$). Statistical significance was evaluated at 0.05 alpha level with GraphPadPrism, version 8.3.1, Mac. Source data are provided in the Source Data file.

In comparison to the only other analyses of CRY1 function in humans, CRY1 appears to harbor distinct activities in carcinomas versus sarcomas. In osteosarcoma cells, CRY1 bound primarily to intergenic regions (49.3%) followed by intronic (28.4%) and promoter regions (7.5%)[19]. Observations in PCa proved quite distinct, in which CRY1 preferably bound to promoter (37.9%) and intronic regions (32.6%), with only a minority in intergenic regions (17.1%) (Fig. 2b). While the sarcoma cells studied do

harbor a circadian clock, a rather small number of clock-controlled genes oscillated, and with low amplitude[19]. Herein, the same core circadian clock genes that are rhythmically controlled by CRY1 in osteosarcoma cells are also bound by CRY1 in PCa (i.e., CRY1, CRY2, PER2, and PER3) (Fig. 2a) showing concordance of binding across models regardless of circadian synchronization. Notably, analyses of the osteosarcoma dataset revealed conservation of at least one CRY1 binding site of

putative impact on HR factor regulation (*RAD51B*). The importance of CRY1 on DNA repair in osteosarcomas has yet to be studied. On balance, data herein reveal entirely new, protumorigenic functions for CRY1, beyond circadian control.

Pathway analysis, functional molecular studies, and biological validation herein revealed that CRY1 governs a discrete network of transcriptional programs of cancer relevance, including regulation of DNA replication, cell cycle control, and multiple DNA repair processes. While there is some limited precedent for CRY1 in influencing cell cycle regulation[10,48,49], the role of CRY1 in transcriptional regulation of double-strand DNA repair regulation was unexpected. Functional assessment herein demonstrated that CRY1 regulates DNA DSB repair in a cascading, temporal fashion inducing first the sensors/mediators of HR followed by induction of HR effectors. Briefly, in response to DNA damage, CRY1 is stabilized (Fig. 4f, Supplementary Fig. 4c–d) and directly binds to key HR factors in a systematic manner to enhance DSB repair (Fig. 7b, Supplementary Fig. 7c). Within 30-min post-radiation, CRY1 binds to *MRE11A*, *RAD50*, *ATM*, and *RAD51*; subsequently, CRY1 binds to regulatory foci of the downstream HR factors *XRCC3* and *POLD2*. Thus, CRY1 orchestrates the DNA repair process through coordinated, temporal control of HR factor expression, and subsequent functional studies demonstrated that CRY1 is necessary for efficient repair (Figs. 6–7). Conversely, CRY1 induction and stabilization, as observed in human disease, enhances the HR process and is strongly associated with poor outcome (Fig. 7e). As CRY1 harbors no independent transcriptional transactivation domain, the underpinning mechanisms of CRY1 function likely occur through modulation of multiple oncogenic transcriptional factors. As described in the previous section (Fig. 2c–d, Supplementary Fig. 2d–e), binding enrichment was observed for a large number of oncogenic transcription factors of established PCa relevance, including ETS factors, basic helix-loop-helix factors, forkhead factors, and zinc finger components. Additionally, de novo enrichment analysis indicated that the most significantly enriched motif for CRY1 binding was highly homologous for NPAS, which belongs to a family of transcription factors with variable activation or repression domains that can heterodimerize or form complexes capable of DNA binding and target gene regulation[23,50]. Future studies will be directed at determining which of these factors most influence the protumorigenic functions of CRY1. Furthermore, CTCF (CCCTC-binding factor) identified here as enriched near sites of CRY1 binding has been previously shown to modulate HR-mediated DNA repair through regulation of MRE11 and recruitment of CtIP, impacting double-strand resection and overall HR efficiency[51]. In sum, CRY1 has a unique binding pattern, strong potential for co-regulation with key oncogenic factors to impact transcriptional control, and functions beyond canonical circadian regulation to impact DNA repair and ultimately stimulate stability and growth.

Critically, findings herein strongly link tumor-specific CRY1 induction with poor outcome and altered DNA repair processes. Data demonstrate that CRY1 can be induced through either amplification or by active AR signaling wherein androgen stimulation leads to AR directly binding to the CRY1 locus to induce expression (Fig. 1a–b, Supplementary Fig. 1a). AR is a key driver of PCa, and this study identifies another avenue to promote disease progression through regulation of CRY1, which further enhances DDR to promote genomic instability and tumor growth. While the mechanisms by which AR specifically binds to the CRY1 locus and induces CRY1 expression remain undefined, these findings identify a target of AR-regulation that promotes cancer phenotypes. Like AR itself, CRY1 is also amplified in a subset of PCa, and these amplification events are associated with poor outcome (Fig. 1). The impact of the CRY1 amplicon in

tumor behavior is a fertile area for future investigation, as co-amplification in this region of 12q23 is frequently observed in concert with adjacent genes of potential relevance to cancer. Specifically, *PARPBP* (DDR factor suppressing replication stress), *ELK3* (ETS factor), *POLE* (DNA repair factor), and *ROCK1/2* (Rho-associated kinases involved cell adhesion, motility, and proliferation)[52–55] are found in chromosomal proximity to the *CRY1* locus. In SU2C/PCF Dream Team cohort[56], CRY1 amplification was present in 19 tumors, of which co-amplification occurred with PARPBP (84%), ELK3 (68%), POLE (68%), ROCK1 (10.5%), and ROCK2 (10.5%). The contribution of these co-amplified events on CRY1-dependent DNA repair will be of importance to discern. Furthermore, while the present study established CRY1 as an upstream effector of HR, genome-wide analyses nominated additional DDR pathways regulated by CRY1 including NHEJ. Thus, it will be worthwhile to further explore the role of CRY1 in shifting DNA repair competencies of additional pathways.

Finally, studies here underscore the overall clinical relevance of CRY1 alterations in cancer. Findings here revealed a strong association between androgen induced CRY1 and poor outcome. These observations are critical, as androgen depletion is standard of care for combination with radiotherapy in locally advanced PCa to suppress AR activity, which impairs DNA repair[57–59]. Furthermore, CRY1 expression strongly correlates with HR factors identified herein as CRY1 regulated (*ATM*, *MRE11A*, and *RAD50*), reinforcing the observation that CRY1 directly regulates HR gene expression to mediate DNA repair. Thus, it will be intriguing to examine the effect of hormone- and radiotherapy on tumors with high CRY1 expression. The role of CRY1 in influencing the response to PARP1/2 inhibitors should also be explored; current clinical testing illustrated the benefit of targeting DNA repair in CRPC via PARP1/2 inhibitors[60,61], and PARP1/2 inhibitors have also been shown to result in improved responses in combination with AR-suppressing therapeutics, irrespective of HR mutational status[62]. Given that CRY1 is AR-regulated and critical for HR-mediated DNA repair, CRY1 status may provide insight into tumors that may respond to PARP1/2 and/or AR suppression. Determining mechanisms to directly antagonize CRY1 function in the clinical setting may also be an important next avenue of investigation. Notably, the FBXL3 E3 ligase is known to prompt CRY1 degradation downstream of AMPK[63–65], and AMPK activators are currently in clinical development[66]. Finally, the concept of chronotherapy can also be considered, which may provide a mechanism to control CRY1 expression. While unstudied in PCa, there is significant evidence in other solid tumor types that adjusting diurnal timing for radiotherapy can alter outcomes[10,67–70]. Irrespective of mechanism employed, the data herein provide the foundation to develop strategies for thwarting AR-mediated CRY1 expression, DNA damage-mediated CRY1 stabilization and/or CRY1 function in PCa as a novel strategy for therapeutic intervention.

Taken together, studies herein reveal fundamental new knowledge of CRY1 function in human malignancy, wherein CRY1 expression is specifically induced in PCa progression and associated with poor outcome. Molecular interrogation and biological assessment of CRY1 function suggest a paradigm shift by revealing a novel mechanism of action in response to genotoxic insult and illuminate a cascading, temporal regulation of HR factors necessary to elicit repair and promote CRPC growth. The present findings underscore the importance of discerning cancer-promoting factors beyond canonical function by yielding critical insight into androgen-regulated CRY1 function and a novel role in DNA repair while nominating potentially impactful therapeutic targets to enhance patient outcome of this lethal disease.

## Methods

**Cell culture and reagents**. C4-2, 22Rv1, and LNCaP cells were purchased from ATCC, authenticated by ATCC, and tested for mycoplasma upon thawing of cells. All C4-2 and LNCaP derived cell lines were cultured and maintained in Improved Minimum Essential Medium (IMEM) (Thermo Fisher Scientific, 10024CV) supplemented with 5% FBS (fetal bovine serum, heat inactivated), 1% L-glutamine (2 mmol/l), and 1% penicillin-streptomycin (100 units/ml). All 22Rv1-derived cell lines were cultured and maintained in Dulbecco's Modified Eagle Medium (DMEM) (Thermo Fisher Scientific, 10017CV) supplemented with 10% FBS, 1% L-glutamine, and 1% penicillin-streptomycin. All cells were cultured at 37 °C at 5% $CO_2$.

**Generation of doxycycline-inducible cell lines**. For generation of inducible (shCON and shCRY1) cell lines, C4-2 and 22Rv1 cells were transduced with SMARTvector Human Inducible nontargeting mCMV-TurboGFP control shRNA or CRY1 shRNA (Dhamacon V3SH11252-225276283) lentiviral vectors. Transduced cells underwent at least three rounds of antibiotic selection with puromycin. These newly generated cell lines were deemed C4-2 shCON and 22Rv1 shCON for the control doxycycline-inducible shRNA cell models and C4-2 shCRY1 and 22Rv1-shCRY1 for the doxycycline-inducible CRY1 shRNA knockdown cell models.

**Chromatin immunoprecipitation (ChIP)-sequencing**. C4-2 cells were plated in hormone-proficient media. ChIP was performed as previously described[15]. Briefly, cells were cross-linked with 1% fresh formaldehyde for 10 mins at room temperature. Chromatin was sheared to 200-700 bp using Diaganode Ultrasonicator for 30 cycles (30 s on, 30 s off). CRY1 antibody was obtained from Dr. Michael Brunner and generated as described[19]. The ChIP-Seq libraries were constructed using the Swift BioSciences ACCEL-NGS 2 s Plus DNA Library kit with ~10 ng of ChIP DNA. NextSeq 500 sequencer from Illumina was utilized to sequence samples at the TJU Sidney Kimmel Cancer Sequencing Core Facility. ChIP-Seq data have been deposited in the GEO repository under the accession number GSE144960. Supplementary Table 1 describes the primers utilized to validate CRY1 binding with ChIP-qPCR.

**ChIP-sequencing analysis**. FASTQ files were assessed for quality using FASTQC v0.11.5. Reads were aligned to the human genome reference version hg19 using bowtie2 v2.3.2[71] with default parameters. Peak calling was performed using MACS2 v2.1.1[72] with combined replicates, utilizing a q < 0.05 cutoff. ChIP-Seq binding heatmaps and profiles were generated using deepTools v2.5.7[73]. Peak annotation and motif analysis performed using Homer v4.10.3[74] using the parameters indicated.

**Clinical sample ChIP-seq data**. AR ChIP-seq data from normal prostate tissue (n = 8) and primary prostate tumor samples (n = 8) were described first in ref. [75], and merged with samtools. Then, the samtools view -s function was utilized to generate merged files with comparable read numbers. Sample profiles were then plotted with the deepTools plotProfile function[73]. Pomerantz et al.[16] data (previously aligned and normalized bigwig files) were downloaded and analyzed in their published format, with deepTools used for visualization. Patient description for the Netherlands cohort is detailed in the Table 1 below. The description for the Pomerantz et al.[16] dataset can be found in the Supplementary Table 1 of the study[16].

**Patient cohort description**. Tissue collected for the JHMI (Johns Hopkins) cohort of patients who underwent RP (radical prostatectomy) between 1992–2010 at John Hopkins Hospital with median follow-up of 108 months for the metastasis endpoint. These patients were selected as a retrospective case-cohort design study for men who underwent radical prostatectomy at high risk and received no additional therapy till the onset of metastasis. More details about sampling and patients are found in the Ross et al., European Urology, 2016 study [PMID 26058959]. Additionally, the Decipher cohort is a prospectively collected cohort as part of the routine clinical use of the Decipher test. Patients in this cohort have not reached the metastasis endpoint yet, so we are using patients with high Decipher score as a surrogate endpoint for metastasis. The clinical characteristics for the Decipher and JHMI (Johns Hopkins) cohorts can be found in Table 2 below.

**Co-expression analysis**. Correlation analysis was performed using cBioPortal[76,77] utilizing data from the studies indicated.

**RNA-sequencing**. C4-2 shCON and C4-2 shCRY1 cells were treated with doxycycline for 3 days to knockdown expression of CRY1 in biological quadruplicate. Following manufacturer's instructions, RNA was extracted and purified using the miRNeasy kit (QIAGEN). TJU Sidney Kimmel Cancer Sequencing Core Facility performed RNA-sequencing. Briefly, TruSeq Stranded Total RNA Library Prep Gold kit was used to construct the RNA-seq libraries. NextSeq 500 sequencer from Illumina was utilized to sequence samples using single-end 75 bp reads.

**Table 1 Clinical characteristics for Netherlands cohort patient data.**

| Sample | Gleason score | Age | Tumor cells percentage in samples (%) |
|---|---|---|---|
| 1 | 3 + 4 = 7 | 69 | 70 |
| 2 | 4 + 4 = 8 | 73 | 40 |
| 3 | 3 + 4 = 7 | 67 | 30 |
| 4 | 4 + 3 = 7 | 68 | 40 |
| 5 | 4 + 5 = 9 | 67 | 60 |
| 6 | 3 + 4 = 7 | 54 | 65 |
| 7 | 3 + 4 = 7 | 62 | 70 |
| 8 | 3 + 4 = 7 | 64 | 30 |

**Table 2 Clinical characteristics for decipher and JHMI patient data.**

| Variable | Prospective decipher cohort No. (%); median (IQR) | Johns Hopkins cohort No. (%); median (IQR) |
|---|---|---|
| Total | 5239 (100%) | 355 |
| Age (years) | 65.5 (60, 69.2) | 59 (56,64) |
| PSA at diagnosis (ng/mL) | 6.5 (4.8, 9.7) | 8.6 (5.7,13.3) |
| <10 ng/mL | 1886 (36%) | 209 (58.9%) |
| 10-20 ng/mL | 441 (8.4%) | 111 (31.2%) |
| >20 ng/mL | 166 (3.1%) | 34 (9.6%) |
| Gleason grade group | | |
| 1 | 271 (5.1%) | 7 (2%) |
| 2 | 1769 (33.7%) | 150 (42.2%) |
| 3 | 1209 (23%) | 66 (18.6%) |
| 4 | 396 (7.5%) | 35 (9.9%) |
| 5 | 554 (10.5%) | 97 (27.3%) |
| PSM | | |
| Present | 2099 (40%) | 102 (28.7%) |
| EPE | | |
| Present | 2092 (40%) | 238 (67%) |
| SVI | | |
| Present | 781 (15%) | 86 (24.2%) |
| LNI | | |
| Positive | 195 (3.7%) | 62 (17.5%) |
| Metastasis outcome | NA | 127 (35.8%) |
| High genomic risk (Decipher) | 1476 (28%) | 46 (12.9%) |
| Median follow-up (months) | 48 [36–54] | 108 [72-144] |

**RNA-sequencing analysis**. FASTQ files were aligned using STAR v2.5.2a[78–80] against the human genome (hg19). Read counts for each gene were generated using featureCounts[81] utilizing Ensembl as reference gene annotation set. Differential gene expression data were generated using DESeq2 v1.12.4[82]. Gene set enrichment analysis (GSEA) was performed using HALLMARKS and KEGG gene signatures from the Molecular Signature Database[83]. RNA-seq data have been deposited in the GEO repository under the accession number GSE144961.

**Flow cytometry**. All C4-2- and 22Rv1-derived cells were plated at equal densities in hormone-proficient media. Once all treatments were completed, cells were incubated with BrdU (1:1000) for 2 h prior to harvesting. Cells were fixed and processed as previously described[84]. At least 10,000 events per sample were assessed. Analysis was performed using InCyte software (Guava) for cell cycle profile with BrdU incorporation and PI (propidium iodide).

**Gene expression analysis**. All C4-2- and 22Rv1-derived cells were plated at equal densities in hormone-proficient media. Trizol (Invitrogen) was used to isolate RNA and SuperScript VILO (Invitrogen) was used to generate cDNA following manufacturer's instructions. PowerSybr (Fischer Scientific 43-676-59) and the ABI

StepOne Real-Time PCR system were utilized in accordance with manufacturer's specifications to perform quantitative PCR (qPCR) analyses. The primers used are depicted in Supplementary Table 2.

**Immunoblotting**. All C4-2- and 22Rv1-derived cells were plated at equal densities in hormone-proficient media. Generation of cell lysates was described previously[6]. Forty to fifty microgram of lysate was resolved by SDS-PAGE, transferred to PVDF (polyvinylidene fluoride) membrane, and analyzed using the following antibodies at 1:1000 dilution—CRY1 (Bethyl A302-614A), ATM (Cell Signaling Technology (CST) 2873), phospho-ATM (Ser1981) (CST 5883 S), CHK2 (Bethyl A300-619A), phospho-CHK2 (Thr68) (CST 2661 T), MRE11 (CST 8344 T), RAD50 (CST 8344 T), RAD51 (Abcam ab63801), XRCC3 (Novus NB100-165), and Vinculin (Sigma–Aldrich V9264).

**Proliferation assays**. All C4-2- and 22Rv1-derived cells were plated at equal densities in hormone-proficient media. Cells were treated with either IR (irradiation), CRY1 activator (KL001, Sigma–Aldrich SML1032), or doxycycline, which was refreshed every 48 h. Cell number was quantified using the Quanti-IT Pico Green dsDNA assay kit (Thermo Fisher) at the indicated times of treatment.

**Xenograft analysis**. C4-2 cells were resuspended in 100 µL of 50% Matrigel (BD Biosciences) and saline mixture followed by subcutaneous injection in athymic nude mice (age at least 6 weeks old). Once tumors reached 150 mm$^3$, treatment of 4 Gy IR was initiated. Mice were sacrificed at 20–30 days post-treatment, and tumors were harvested to assess for CRY1 staining. All the mice used in this study were male. No mice lost more than 5% of their body weight throughout the duration of the study. The Institutional Animal Care and Use Committee (IACUC) at Thomas Jefferson University approved all protocols for this study. Importantly, this study has complied with all relevant ethical regulations for animal testing and research.

**Patient Derived Explant (PDE)**. Deidentified prostate tissues (matched non-neoplastic and tumor) were utilized as ex vivo PDE cultures as previously described[7,39–41,85]. TJU's Institutional Review Board has reviewed this protocol and deemed this research to follow federal regulations [45 CFR 46.102(f)]. PDE cultures were treated with 0.5 Gy IR and harvested after 48 h for immunohistochemistry (IHC) analyses.

**Immunohistochemistry (IHC)**. For histological analysis from xenograft and PDE tissue, FFPE sections were stained with CRY1 (1:250) (LifeSpan Biosciences LS-B6955) using standard techniques previously described[6].

**Homologous recombination (HR) activity assay**. U2OS-DR-GFP cells are a modified osteosarcoma cell line that were generated by Dr. Jasin and were utilized to assess HR activity as previously described[43,44]. Dr. Roger A. Greenberg (University of Pennsylvania) provided these cells that were utilized in this study. These cell lines Cells were transfected with siCRY1 or siBRCA1 using Dharmafect 4 reagent following manufacturer's instructions. 48 h post-transfection, ISce1 plasmid was transfected into cells to induce DNA breaks. Cells were treated with ATM inhibitor (KU-55933, Sigma–Aldrich SML1109) or CRY1 Activator (KL001, Sigma–Aldrich SML1032) for the last 16 h of the assay. Cells were harvested and GFP positive cells were quantified via flow cytometry.

**Immunofluorescence (IF)**. Cells were plated at equal densities on poly-L-lysine coated coverslips and treated as indicated. The antibodies used were γH2AX phospho-S139 (EMD Millipore 16-202 A), RAD51 (Abcam ab133534), and CRY1 (Thermo Fisher Scientific PA1-527) at 1:500 dilution. IF assay was performed as described[7]. Foci were imaged utilizing Zeiss Cell Discoverer Confocal Microscope at 40X magnification with at least five fields for each replicate. Fiji image software was utilized to quantify foci formation and compared to control samples.

**Statistics**. All experiments were performed in technical triplicate with at least three biological replicates per condition. Data are displayed as mean ± standard error of the mean (SEM). Statistical significance ($p < 0.05$) was determined using Student's $t$-test, one-way ANOVA, and two-way ANOVA on GraphPad Prism Software (version 8.3.1) as appropriate and indicated in applicable figure legends.

**Study approval**. The use of patient and clinical material was approved by the ethical committees from each of the following institutes: the Sidney Kimmel Cancer Center at Thomas Jefferson University (Pennsylvania, USA), the Department of Radiation Oncology at the University of California at San Francisco (California, USA), and Division of Oncogenomics in the Oncode Institute and the Netherlands Cancer Institute (Amsterdam, The Netherlands).

**Reporting summary**. Further information on research design is available in the Nature Research Reporting Summary linked to this article.

## Data availability
The datasets generated during the current study have been deposited in public repositories. ChIP-Seq data have been deposited in the NCBI Gene Expression Omnibus (GEO) with the accession code GSE144960. RNA-Seq data have been deposited in the NCBI GEO with the accession code GSE144961. Molecular Signature Database (MSigDB) was utilized for pathway analyses. Source data are provided with this paper.

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

## Acknowledgements

This work was supported by a Young Investigator Award and Challenge Award from the Prostate Cancer Foundation (to A.A.S. and K.E.K., respectively), NCI grant F99CA212225 (J.J. M.), NCI R01-CA182569 (K.E.K.), Wilbert Zwart (KWF Dutch Cancer Society), the Sidney Kimmel Cancer Center (SKCC) Support Grant (5P30CA056036); and the Cancer Genomics and Translational Research/Pathology core services at SKCC. Additionally, we would like to thank Dr. R. Greenberg (University of Pennsylvania) for providing the U2OS-DR-GFP cells, Dr. E. Grabocka (Thomas Jefferson University) for use of the Zeiss Cell Discoverer Confocal Microscope, and Dr. M. Lazar for his expertise and guidance on circadian regulation. We would like to also thank Ms. Elizabeth Scade for her assistance with the artwork to generate original images utilized in the models in this manuscript. The collaborators played a role in the design of the study; data collection; analysis; interpretation; and review and approval of the manuscript. We are grateful for each individual support and input. Although unrelated to the study, Dr. Knudsen reports consulting/advisory relationships with CellCentric and Genentech, and has served on ad hoc advisory panels for Sanofi, Celgene, and Atrin.

## Author contributions

Conceptualization, A.A.S. and K.E.K; Methodology, A.A.S., C.M.M., J.J.M., and K.E.K.; Software, C.M.M.; Validation, A.A.S.; Formal Analysis, A.A.S., C.M.M, and W.Z.; Investigation, A.A.S., M.A., T.M.S., Y.Z., A.B., and M.J.S.; Resources, A.A.S., A.S., M.B., F.Y.F., W.Z., and K.E.K.; Writing—Original Draft, A.A.S. and K.E.K; Writing—Review and Editing, A.A.S., C.M.M., J.J.M., N.G., A.C.M., S.N.C., P.G., E.D., T.S.L., I.A.V., M.J.S., and K.E.K; Visualization, A.A.S.; Funding Acquisition, A.A.S. and K.E.K.

## Competing interests

The following are disclosures for K.E.K.: Research Support (Celgene, CellCentric) and Consultant (CellCentric, Sanofi, Atrin, Celgene, Janssen, Genentech). No disclosures for the other authors.
