## [Peer Review File · Nature Communications]

REVIEWER COMMENTS

Reviewer #1 (Remarks to the Author):

In this study by Shafi, et al, the authors present an elegant analysis for the role of CRY1 in DNA repair modulation utilizing human and experimental studies. They find that the circadian factor CRY1 is a tumor specific regulator of DNA repair, and is associated with poor outcome in prostate cancer. Comments on specific aspects of the paper are below:

Introduction:

1) there has been prior work evaluating circadian rhythm disruption and cancer risk and progression, including in prostate cancer that would add to the story.

Methods:

1) Please provide additional details on some of the human studies included. For example, the JHMI cohort, how were the patients selected for that study? Is the expression in tumor, adjacent-normal or true normal? Is the tissue prostate or any site among those with prostate cancer?

2) Is the expression similar in normal adjacent and tumor in the same person? Is expression/mutations just altered because of the tumorigenesis process?

3) Are any of the outcomes (Fig 1D) adjusted for age? Age is associated with survival and has been shown to be associated with other markers of circadian disruption.

4) Supp Fig 1A shows AR binding sites on CRY1 at different cell cycle phases – does CRY1 expression differ by cell phase or time of collection/assessment?

5) Supp Fig 1B. shows all CRY1 in prostate are amplifications, while in uterine are all mutations and in other cancer types, varying. Please comment on potential reasons for this and/or if this adds anything to our understanding of the role of CRY1 in prostate specifically. Also, the alteration frequency is low for most cancer types (<4%) – how does this compare to other genes?

6) Some caution is warranted about interpreting Fig 1D – the curves start to diverge really around 50-100 months and the comparison is down to 12 and 11 people in the low vs. high group.

7) Please clarify the cohorts/numbers for Fig 1D, 1E and Supp 1D – Supp 1D is referred to in relation to fig 1D (JHMI cohort) and Fig 1E (Decipher cohort).

8) Figure 1E – can the authors put a correlation coefficient on the correlation between CRY1 and Decipher score?

9) Fig 7D shows correlation between DDR and CRY1 genes. It would be interested to see the cross-tab association with outcomes – ie highATM/highCRY1 vs lowATM/lowCRY1 vs. highATM/lowCRY1 vs. lowATM/highCRY1. Is this correlation also seen in localized tumor setting?

Discussion:

A bit more caution should be applied to the human findings of an association between CRY1 expression and poor outcome given the numbers, lack of detailed descriptions of the study populations, and unadjusted estimates.

Reviewer #2 (Remarks to the Author):

The manuscript by Shafi et al describes for the first time that the circadian factor CRY1 is an androgen regulator gene and an effector of homologous recombination (HR) in prostate cancer. ChIPseq analysis revealed that CRY1 binds to the promoter HR gene to regulate HR-mediated DNA damage response as a sensor and a mediator. Particularly, they found that CRY1 is stabilized in response to genotoxic insult induce rapid repair of DNA double strand break by directly regulating HR gene expression and supporting cell growth. Overall data presented here linking CRY1 to androgen receptor and to the regulation of DNA repair in prostate cancer are novel.

Specific comments:

The authors showed that AR binds 5 different regions within CRY1 locus in LNCaP and VCaP, please clarify if these experiments were done in the presence or absence of androgen. It will be interesting to show the tracks from -/+ hormone.

The AR ChIPseq data from patients are interesting and further support the in vitro data that AR binds to CRY1 locus. What is the nature of the disease in these patients, are they naïve or treated? Provide more details (reference to fig 1C).

How the genomic risk of metastasis was defined in figures 1E, S1D?

CRY1 cistrome analysis revealed that CRY1 binds to a small fraction of genes involved in the circadian function but it was found to bind to genes involved in growth factors, DNA repair and metabolic processes. Are these observations a result of hormone or specific CRY1 function in prostate cancer? A comparison of CRY1 cistrome in prostate cancer cells treated and no treated with hormone will be informative.

Data presented in this manuscript showed that CRY1 is more stabilized by DNA damage compared to known CRY1 activator while the authors concluded that "the findings strongly support the conclusion that CRY1 is activated by stabilization after DNA damage, similar with the known CRY1 activator KL001. It will be important to compare DNA damage insult and KL001 in the same blot. There is a difference in the effect of KL001 on the stability of CRY1 in C4-2 versus 22RV, any explanation?

What is the rationale of using MG132 since it is not specific to CRY1?

How the DNA damage affect CRY1 cistrome? Is the effect of DNA damage on CRY1 cistrome similar to effect of hormone treatment?

Minor:

Labelling of figure 4 has to be improved.

Thank you for considering the manuscript #NCOMMS-20-22319-T, “The Circadian Cryptochrome, CRY1, is a Pro-Tumorigenic Factor that Rhythmically Modulates DNA Repair” for publication to *Nature Communications*. We are pleased with the positive response from the reviewers, and their appreciation of the importance of the findings. We have addressed each reviewer comment to completion as described below, which further increased the impact of the study.

Response to Critiques (in order of appearance)

Reviewer #1:

In this study by Shafi, et al., the authors present an elegant analysis for the role of CRY1 in DNA repair modulation utilizing human and experimental studies. They find that the circadian factor CRY1 is a tumor specific regulator of DNA repair, and is associated with poor outcome in prostate cancer. Comments on specific aspects of the paper are below:

Introduction:

1) There has been prior work evaluating circadian rhythm disruption and cancer risk and progression, including in prostate cancer that would add to the story.

Response: We appreciate this important point and have added findings from key epidemiological studies on circadian disruption and cancer incidence in the Introduction section on pages 3-4. Thank you for this valuable suggestion.

Methods:

1) Please provide additional details on some of the human studies included. For example, the JHMI cohort, how were the patients selected for that study? Is the expression in tumor, adjacent-normal or true normal? Is the tissue prostate or any site among those with prostate cancer?

Response: This is an important point, and we have added the requested information on pages 20-22 in the Methods section. Tissue collected for the JHMI (Johns Hopkins Medical Institute) cohort are tumor tissues from the prostate after surgery. These patients were selected as a retrospective case-cohort design study for men who underwent radical prostatectomy and received no additional therapy until the onset of metastasis. More details about sampling and patients are found in this paper [PMID 26058959, Ross *et al*, *European Urology*, 2016]. Below is the clinical characteristic table for the Decipher and JHMI cohort now found in the revised Methods section on pages 21-22.

Clinical Characteristics for Decipher and JHMI Cohort Patient Data

Variable	Prospective Decipher Cohort No. (%); Median (IQR)	Johns Hopkins Cohort No. (%); Median (IQR)
Total	5,239 (100%)	355
Age (years)	65.5 (60, 69.2)	59 (56,64)
PSA at diagnosis (ng/mL)	6.5 (4.8, 9.7)	8.6(5.7,13.3)
<10 ng/mL	1886 (36%)	209 (58.9%)
10-20 ng/mL	441 (8.4%)	111 (31.2%)
>20 ng/mL	166 (3.1%)	34 (9.6%)
Gleason Grade group		
1	271 (5.1%)	7 (2%)
2	1769 (33.7%)	150 (42.2%)
3	1209 (23%)	66 (18.6%)
4	396 (7.5%)	35 (9.9%)

5	554 (10.5%)	97 (27.3%)
PSM		
Present	2099 (40%)	102 (28.7%)
EPE		
Present	2092 (40%)	238 (67%)
SVI		
Present	781 (15%)	86 (24.2%)
LNI		
Positive	195 (3.7%)	62 (17.5%)
Metastasis outcome	NA	127 (35.8%)
High genomic risk	1476 (28%)	46 (12.9%)
(Decipher)		
Median follow-up	48[36-54]	108 [72-144]
(months)		

2) Is the expression similar in normal adjacent and tumor in the same person? Is expression/mutations just altered because of the tumorigenesis process?

Response: Unfortunately, this patient cohort does not contain adjacent normal and tumor tissue from same patient, which would be needed to address this question. While such analyses will be the focus of future studies, the present data show that CRY1 expression is strongly associated with metastasis and poor outcome. Additionally, the present study implicates CRY1 induction (as achieved by AR signaling and/or amplification of the *CRY1* locus) as an effector of disease progression. Expression changes and increased mutational frequency are often found with the tumorigenesis process. Accordingly, for CRY1, there is increased frequency of CRY1 alterations from primary to metastatic disease as seen below from cBioPortal analyses of primary and metastatic prostate cancer patient cohorts. Analyses of frequency alterations of the core circadian clock genes (*CLOCK*, *BMAL*, *CRY1*, *CRY2*, *PER1*, and *PER2*) indicated that CRY1 was the second most altered circadian gene in primary disease and importantly the most altered circadian gene in metastatic disease, in which CRY1 is predominantly amplified. These data are now added as the new Supplemental Figure 1E with a revised figure legend on pages 35-36 and discussed in the Results section on page 5.

E.

Supplemental Figure 1. Altered CRY1 expression is associated with different types of cancer. E. Frequency of core circadian clock gene alterations (i.e. amplifications, mutations, and/or deletions) in primary and metastatic PCa datasets from cBioPortal.

3) Are any of the outcomes (Fig 1D) adjusted for age? Age is associated with survival and has been shown to be associated with other markers of circadian disruption.

Response: We appreciate the reviewer bringing attention to this additional analysis for comparison. Interestingly, even when adjusting for age, high CRY1 remained an independent prognostic variable (HR 1.56 with a 95%CI [1.04-2.34], $p=0.029$), further strengthening the importance of CRY1 functioning as a pro-tumorigenic factor in prostate cancer. This additional comparison and analysis from the JHMI study has been added to the Results section on page 5.

4) Supp Fig 1A shows AR binding sites on CRY1 at different cell cycle phases – does CRY1 expression differ by cell phase or time of collection/assessment?

Response: This is an excellent question. When analyzing mRNA expression of CRY1 across the different cell cycle phases, there is no significant change in expression as depicted below. These findings further complement the previous conclusion that CRY1 levels are not altered in a circadian manner in prostate cancer cells and extend this result to show that CRY1 is also not cell cycle regulated. These data are now as the new Supplemental Figure 1B with a revised figure legend on pages 35-36 and discussed in the Results section on page 5.

B.

Supplemental Figure 1. Altered CRY1 expression is associated with different types of cancer. B. LNCaP cells were fixed in different phases of the cell cycle (i.e. early G1, late G1, early S, mid S, and G2-M) and analyzed for gene expression in the McNair *et al*, 2017. CRY1 mRNA expression across the phases of the cell cycle was analyzed to assess any impact of cell cycle phase on gene expression.

5) Supp Fig 1B. shows all CRY1 in prostate are amplifications, while in uterine are all mutations and in other cancer types, varying. Please comment on potential reasons for this and/or if this adds anything to our understanding of the role of CRY1 in prostate specifically. Also, the alteration frequency is low for most cancer types (<4%) – how does this compare to other genes?

Response: We performed additional analyses of patient cohorts to address this important question. First, to compare other core circadian clock genes, distribution of *CLOCK*, *BMAL*, *CRY2*, *PER1*, and *PER2* alongside *CRY1* was analyzed in primary and metastatic disease. As shown in primary disease (i.e. TCGA dataset, Cell 2015), *CRY1* (18.0%) was the second most altered circadian gene in PCa following *PER1* (28.2%). Interestingly, here *CRY1* is predominantly amplified. In metastatic disease, *CRY1* (37.7%) was the most significantly altered circadian gene followed (37.7%), by *CRY2* (31.4%). Although less than in primary PCa disease, *CRY1* was still mostly amplified followed by gene deletion. These data are now

added as the new Supplemental Figure 1E with a revised figure legend on pages 35-36 and discussed in the Results section on page 5. Interestingly, there was no evidence of somatic *CRY1* mutations in PCa disease, distinct from observations in uterine, lung, and melanoma cancers. Importantly, in uterine and melanoma, exogenous circadian clock synchrony is sustained, in which core circadian genes are known to oscillate in a rhythmic manner. However, in PCa cells there is a failure of the cell models to *in vitro* circadian sync as described in the Results section of the manuscript on page 6. Future studies will address the function of *CRY1* and additional core circadian genes in different disease models to understand the importance of the *CRY1* somatic mutations on disease progression.

E.

Supplemental Figure 1. Altered *CRY1* expression is associated with different types of cancer. E. Frequency of core circadian clock gene alterations (i.e. amplifications, mutations, and/or deletions) in primary and metastatic PCa datasets from cBioPortal.

To further address the reviewer query, *CRY1* alteration frequency was compared to other genes in PCa, including *AR*, *AR* targets (*KLK3* (i.e. PSA), *FKBP5*), *p53*, and *PTEN*. These additional analyses show that genes frequently amplified (i.e. *AR*, *KLK3*, *FKBP5*) in PCa are to the similar frequency as *CRY1* with frequency alterations from 1.2%-1.8%. Interestingly, the greater percentage of frequency changes are seen in genes with deep deletions as depicted in *TP53* (8%) and *PTEN* (17%). This suggests that the *CRY1* alteration frequency is similar to other key PCa gene amplifications found in patients. These data are now added as the new Supplemental Figure 1F with a revised figure legend on pages 35-36 and discussed in the Results section on page 5.

F.

Supplemental Figure 1. Altered CRY1 expression is associated with different types of cancer. F. Gene alteration frequencies of CRY1, AR, KLK3, FKBP5, TP53, and PTEN were assessed from the TCGA (Cell 2015) primary PCa dataset publicly available on cBioportal. Amplifications, mutations, and/or deletions were assessed.

6) Some caution is warranted about interpreting Fig 1D – the curves start to diverge really around 50-100 months and the comparison is down to 12 and 11 people in the low vs. high group.

Response: We thank the reviewer for this important caution in interpretation. Statistical analysis performed on the overall study for survival differences between the “low CRY1” and “high CRY1” cohorts indicated a significant difference in overall metastasis-free survival. As is now described in the revised text, the y-axis is metastasis after RP (radical prostatectomy). Thus, it is not expected that patients develop metastasis within 40 months of surgery. Most patients develop metastasis after 5-7 years (60-84 months) and that is where the separation between groups was observed. We have added description of this cohort to the Methods section on pages 21-22 to provide more clarity and modified the interpretation in the Results section on page 5.

7) Please clarify the cohorts/numbers for Fig 1D, 1E and Supp 1D – Supp 1D is referred to in relation to fig 1D (JHMI cohort) and Fig 1E (Decipher cohort).

Response: We apologize that this important description was not added initially to the manuscript. JHMI (Johns Hopkins Medical Institute) cohort is a retrospective cohort of patients who underwent RP (radical prostatectomy) between 1992-2010 at John Hopkins Hospital with median follow-up of 108 months for the metastasis endpoint. The Decipher cohort is a prospectively collected cohort as part of the routine clinical use of the Decipher test. Patients in this cohort have not reached the metastasis endpoint yet, so patients with high Decipher score were used as a surrogate endpoint for metastasis. This description has been added to the Methods section on pages 21-22.

8) Figure 1E – can the authors put a correlation coefficient on the correlation between CRY1 and Decipher score?

Response: This important statistical analysis has now been added to the Results section on page 5 and to Figure 1E. The correlation coefficient is 0.07 ($p=1e-7$).

9) Fig 7D shows correlation between DDR and CRY1 genes. It would be interested to see the cross-tab association with outcomes – ie highATM/highCRY1 vs lowATM/lowCRY1 vs. highATM/lowCRY1 vs. lowATM/highCRY1. Is this correlation also seen in localized tumor setting?

Response: This is an excellent query. While CRY1 correlated with ATM in the MSKCC (Cancer Cell 2010) and Board/Cornell (Nature Genetics 2012) data sets, unfortunately, these cohorts cannot be subcategorized into “high” and “low” for ATM and/or CRY1. Such analysis will be of interest for future studies. Interestingly, the localized tumor datasets available for analyses on cBioPortal did not show significant correlation while the metastatic cohorts did show strong correlation supporting the conclusion that CRY1 is a pro-tumorigenic factor directly mediating homologous recombination (HR)-mediated DNA repair.

Discussion:

1) A bit more caution should be applied to the human findings of an association between CRY1 expression and poor outcome given the numbers, lack of detailed descriptions of the study populations, and unadjusted estimates.

Response: We thank the reviewer for bringing forth this point, and as a result additional details of the patient cohorts (*i.e.* JHMI, Decipher, Netherlands cohort, and Pomerantz data) were added in the Methods section of the manuscript on pages 20-22. Furthermore, additional analysis for the correlation coefficient on the comparison between CRY1 and the Decipher

score was completed and now added to the Results section of the manuscript on page 5 and to Figure 1E (shown above as well), which further strengthened the human data findings on the association between CRY1 and poor outcome. Additionally, the dataset was further analyzed to adjust for age. In these analyses, high CRY1 remained an independent prognostic variable (HR 1.56 with a 95%CI [1.04-2.34], p=0.029) further strengthening the importance of CRY1 functioning as a pro-tumorigenic factor in prostate cancer. This additional comparison and analysis from the JHMI study has been added to the Results section of the manuscript on page 5. Lastly, detail describing the components used to define the genomic risk of metastasis in the Decipher cohort were added. The genomic risk of metastasis is defined based on the Decipher test score which is a strong predictor of metastasis. This description has been added to the Results section on page 5. We thank the reviewer for these outstanding suggestions, which solidified data surrounding the strong association between CRY1 expression and poor outcome.

Reviewer #2:

The manuscript by Shafi et al describes for the first time that the circadian factor CRY1 is an androgen regulator gene and an effector of homologous recombination (HR) in prostate cancer. ChIPseq analysis revealed that CRY1 binds to the promoter HR gene to regulate HR-mediated DNA damage response as a sensor and a mediator. Particularly, they found that CRY1 is stabilized in response to genotoxic insult induce rapid repair of DNA double strand break by directly regulating HR gene expression and supporting cell growth. Overall data presented here linking CRY1 to androgen receptor and to the regulation of DNA repair in prostate cancer are novel.

Specific comments:

The authors showed that AR binds 5 different regions within CRY1 locus in LNCaP and VCaP, please clarify if these experiments were done in the presence or absence of androgen. It will be interesting to show the tracks from +/- hormone.

Response: This is an important query to determine the presence of AR binding on the CRY1 locus in response to hormone stimulation. The current ChIP-Seq studies in LNCaP and VCaP cell models were performed in the presence of hormone (either 1 nM R1881 or 10 nM DHT) for stimulation of AR. Specifically, each study used the following conditions:

a) Barfeld *et al*, 2017 – LNCaP cells stimulated with hormone (16 hours of 1 nM R1881 treatment after 72 hours serum starvation). As described in the Methods section of Barfeld *et al* 2017, this study did not have a vehicle (i.e. no hormone stimulation) AR ChIP-Seq to assess AR binding to the CRY1 locus in the absence of hormone treatment.

b) Takayama *et al* 2018 – LNCaP cells stimulated with hormone (10 nM DHT for 24 hours). As described in the Methods section of Takayama *et al* 2018, this study did not have a vehicle (i.e. no hormone stimulation) AR ChIP-Seq to assess AR binding to the CRY1 locus in the absence of hormone treatment.

c) Asangani *et al* 2014 – VCaP cells stimulated with hormone (10 nM DHT for 12 hours). This study had a vehicle (i.e. no hormone stimulation) AR ChIP-Seq to assess AR binding. The track below shows that AR does not bind to the CRY1 locus in the absence of hormone treatment.

d) Massie *et al* 2011 – VCaP cells stimulated with hormone (1 nM R1881 for 4 hours after serum starvation). This study had a vehicle (i.e. no hormone stimulation) AR ChIP-Seq to assess AR binding. The track below shows that AR does not bind to the CRY1 locus in the absence of hormone treatment.

To address this important point, the tracks depicting AR binding at the CRY1 locus with and without hormone stimulation were added as Supplemental Figure 1A with a revised figure legend on pages 35-36 and discussed in the Results section on pages 4-5. Here it is important to note that both studies examining AR binding in VCaP cells found that androgen stimulation

(via 10 nM DHT or 1 nM R1881) enhanced recruitment and binding of AR to the CRY1 locus at several sites as shown below.

Supplemental Figure 1. Altered CRY1 expression is associated with different types of cancer. A. AR binding sites on CRY1 in PCa data sets of VCaP cells (Asangani *et al.* 2014 and Massie *et al.* 2011).

Thus overall, it is important to note that in a hormone-responsive PCa cell model (VCaP), in the absence of hormone stimulation, there is no AR binding to the CRY1 locus. Importantly, when AR is activated through androgen treatment, there is strong binding of AR at several sites on the CRY1 locus. These observations further support the conclusion that CRY1 is androgen responsive since AR directly bound to the CRY1 locus and increased CRY1 mRNA and protein expression (Figures 1A-B, Supplemental Figure 1A).

The AR ChIPseq data from patients are interesting and further support the *in vitro* data that AR binds to CRY1 locus. What is the nature of the disease in these patients, are they naïve or treated? Provide more details (reference to fig 1C).

Response: This is an important point by the reviewer for further clarification that has now been added in the Methods section on pages 20-21 to provide clarity about AR binding to the CRY1 locus in human patient tissue. Briefly, for the Netherlands Cohort, the patient samples were derived from treatment naïve patients, collected following RP (radical prostatectomy). Below we have added clinical characteristics for this cohort including Gleason score, age, and tumor percentage information (see below). We thank the reviewer for bringing our attention to this important detail to enhance understanding on this novel and translational finding.

Clinical Characteristics for Netherlands Cohort Patient Data

Sample	Gleason score	Age	Tumor cells percentage in samples (%)
1	3+4=7	69	70
2	4+4=8	73	40
3	3+4=7	67	30
4	4+3=7	68	40
5	4+5=9	67	60
6	3+4=7	54	65
7	3+4=7	62	70
8	3+4=7	64	30

For the published Pomerantz et al, 2015 data set for AR binding in PCa patient samples, this cohort analyzed 13 independent PCa samples along with 7 histologically normal samples from areas of fresh-frozen radical prostatectomy (RP) tissue specimens. Description of the patient cohort from this study is referenced in the revised Methods sections on page 20. The authors note in the Pomerantz et al, 2015 study that all the specimens had at least 70% epithelial enrichment. Additionally, 6 cases of tumor samples had matched normal tissue for comparison. Supplemental Table 1 from the study that was published in their manuscript

(Pomerantz et al, 2015) provides more specific details of the tissue samples utilized for AR ChIP-Seq analysis.

How the genomic risk of metastasis was defined in figures 1E, S1D?

Response: We apologize that this important description was sufficiently detailed in the initially submitted manuscript and now have detailed the components used to define the genomic risk of metastasis in the Decipher cohort. The genomic risk of metastasis is defined based on the Decipher test score which is a strong predictor of metastasis. Specifically, the Decipher test is a tissue-based tumor genomic test that predicts the probability of metastasis within 5 years of RP (radical prostatectomy) and provides an independent assessment of tumor aggressiveness (Dalela et al, Rev Urol, 2016). This is additional information that is distinct from that clinically utilized Gleason score or PSA. This description has been added to the Results section on page 5.

CRY1 cistrome analysis revealed that CRY1 binds to a small fraction of genes involved in the circadian function but it was found to bind to genes involved in growth factors, DNA repair and metabolic processes. Are these observations a result of hormone or specific CRY1 function in prostate cancer? A comparison of CRY1 cistrome in prostate cancer cells treated and no treated with hormone will be informative.

Response: This is an important point and we appreciate the reviewer bringing this to our attention. The novel CRY1 ChIP-Seq study performed in this manuscript was done in a clinically relevant hormone-proficient condition. CRY1 function was not examined in a hormone-deficient condition, so we performed ChIP-qPCR analyses on the circadian and DNA repair (i.e. HR specific targets) directly bound by CRY1 in C4-2 cells in hormone-deficient conditions (i.e. CDT (charcoal dextran-treated)) and also in the presence of hormone stimulation (i.e. CDT+DHT (dihydrotestosterone)). Binding of CRY1 was not significantly increased at the CRY1 circadian target (i.e. CRY2) or at DNA repair factors regulated by CRY1 (i.e. MRE11A, ATM, XRCC3, and POLD2) in the presence of hormone stimulation. Statistical analysis utilizing two-way ANOVA indicated CRY1 binding at all the targets depicted with significant over the respective desert region except for MRE11A in the CDT+DHT condition. This suggests that the observed binding of CRY1 was specific to CRY1 and not due to hormone stimulation. These important data solidify the observation that CRY1 is a pro-tumorigenic factor specifically binding to and regulating circadian and DNA repair factors important in disease progression. These data are now added as the new Supplemental Figure 7F with a revised figure legend on page 38 and discussed in the Results section on page 13.

F.

Supplemental Figure 7. CRY1 directly binds to promoters of HR genes and regulates HR gene expression to promote DNA repair. F. CRY1 ChIP qPCR was performed on CRY2,

MRE11A, ATM, XRCC3, POLD2, and a desert region to determine the binding of CRY1 in CDT and CDT+DHT (10 nM, 3 hours) treatment in C4-2 cells. CRY1 binding to these sites is plotted as percent input. n=3, *p<0.05, **p<0.01.

Data presented in this manuscript showed that CRY1 is more stabilized by DNA damage compared to known CRY1 activator while the authors concluded that “the findings strongly support the conclusion that CRY1 is activated by stabilization after DNA damage, similar with the known CRY1 activator KL001. It will be important to compare DNA damage insult and KL001 in the same blot.

Response: We thank the reviewer for this important point and have now assessed CRY1 in response to DNA damage insult (i.e. 5 Gy IR over time) and CRY1 activator (KL001) treatment in the same blot for comparison to directly address this point. As shown below and incorporated into the revised supplemental data as Supplemental Figures 4A-B, data indicate that by 4 hours of IR CRY1 protein expression is stabilized to a similar extent as seen with CRY1 activator (KL001) treatment. These data are now added in the supplemental data as the new Supplemental Figures 4A-B with a revised figure legend on page 37 and discussed in the Results section on page 10. Importantly, this supports the findings in Figure 4 with CRY1 protein expression is stabilized 2-3 fold in response to genotoxic insult as depicted below through ImageJ analysis to quantify protein expression and assess fold change. Additionally, CRY1 activator (KL001) increased protein expression to a similar extent now depicted more clearly in the new Supplemental Figures 4A-B as compared on the same blot. These findings underscore the important conclusion that CRY1 is stabilized by DNA damage and this is to a similar extent as through pharmacological activation of CRY1 via KL001.

Supplemental Figure 4. DNA damage stabilizes CRY1 protein expression. A-B. C4-2 and 22Rv1 cells were treated 5 Gy IR for 0-8 hours and with 10 μ M KL001 for 6 and 24 hours, respectively. Cells were harvested; protein expression of CRY1 & Vinculin was analyzed and quantified using ImageJ.

There is a difference in the effect of KL001 on the stability of CRY1 in C4-2 versus 22RV, any explanation?

Response: As shown below and incorporated into the revised figures as the new Figure 4E, basal CRY1 protein expression is higher in 22Rv1 cells in comparison to C4-2 cells. These data are now added as the new Figure 4E with a revised figure legend on page 34 and discussed in the Results section on page 10. This higher CRY1 protein expression in 22Rv1 cells likely accounts for the attenuated induction of CRY1 protein with activator (KL001) treatment (i.e. 1.9-fold increase). On the other hand, C4-2 cells have lower basal CRY1 expression and the induction of CRY1 protein expression with activator (KL001) treatment is higher (i.e. 3.1-fold increase). In sum, these data support the conclusion that CRY1 is stabilized after DNA damage, similar to that observed with the known CRY1 activator KL001.

E.

Figure 4. DNA damage results in CRY1 stabilization. E. C4-2 & 22Rv1 cells were treated with 10 μ M KL001 for 6 and 24 hours, respectively. A-E. Cells were harvested; protein expression of CRY1 & Vinculin was analyzed and quantified using ImageJ.

What is the rationale of using MG132 since it is not specific to CRY1?

Response: We apologize for the lack of clarity on the rationale for utilizing MG132 to examine CRY1 protein expression. To ensure that all potential avenues of CRY1 degradation were targeted, the broad proteasome inhibitor (MG132) was used to examine the stability of CRY1 and determine whether degradation of CRY1 is proteasome dependent. MG132 functions to reduce the degradation of ubiquitin-conjugated proteins in mammalian cells and is frequently employed to study the stability and degradation of proteins since active ubiquitin tagged protein degradation via proteasomal function is blocked with the treatment of MG132 (Lee *et al*, 1998). This is now clarified in the Results section on page 10.

How the DNA damage affect CRY1 cistrome? Is the effect of DNA damage on CRY1 cistrome similar to effect of hormone treatment?

Response: This is an important point of comparison brought up by the reviewer. Figures 7B and supplemental Figure 7C show that CRY1 directly binds to HR-targets after genotoxic insult. ChIP qPCR studies were implemented to examine the impact of androgen stimulation on CRY1 binding to key DNA repair targets to address this important point from the reviewer. As depicted below, CRY1 ChIP qPCR was performed in C4-2 cells in hormone-deficient conditions (*i.e.* CDT (charcoal dextran treated)) and also in the presence of hormone stimulation (*i.e.* CDT+DHT (dihydrotestosterone)). Statistical analysis utilizing two-way ANOVA indicated CRY1 binding at all the DNA repair targets evaluated with significant binding of CRY1 over the respective desert region except for MRE11A in the CDT+DHT condition. The binding of CRY1 was not significantly increased at DNA repair factors regulated by CRY1 (*i.e.* MRE11A, ATM, XRCC3, and POLD2) in the presence of hormone stimulation more than CDT alone. These data are now added in the supplemental data as the new Supplemental Figure 7F with a revised figure legend on page 38 and discussed in the Results section on page 13. Together, this suggests that the binding of CRY1 observed was specific to CRY1 and not due to hormone stimulation. Moreover, with genotoxic insult, CRY1 binding is enhanced in a rhythmic manner to direct HR-mediated repair as shown in Figure 7B. In conclusion, these important data solidify the observation that CRY1 is a pro-tumorigenic factor specifically binding to and regulating DNA repair factors important in disease progression.

F.

Supplemental Figure 7. CRY1 directly binds to promoters of HR genes and regulates HR gene expression to promote DNA repair. F. CRY1 ChIP qPCR was performed on CRY2, MRE11A, ATM, XRCC3, POLD2, and a desert region to determine the binding of CRY1 in CDT and CDT+DHT (10 nM, 3 hours) treatment in C4-2 cells. CRY1 binding to these sites is plotted as percent input. n=3, *p<0.05, **p<0.01.

Minor:

Labelling of figure 4 has to be improved.

Response: We appreciate this suggestion and have improved the labelling of the western blots to ease understanding of this figure and the data. We hope this makes the representation of the data easier to follow.

REVIEWERS' COMMENTS

Reviewer #1 (Remarks to the Author):

The authors have adequately addressed my concerns.

Reviewer #2 (Remarks to the Author):

the authors responded to all reviewer's comments.

The authors should provide high quality western blot for supplemental figure 4.